# Molecular mechanism and structural basis of small-molecule modulation of the gating of acid-sensing ion channel 1

Yi Liu [1✉], Jichun Ma [2], Renee L. DesJarlais[2], Rebecca Hagan[1], Jason Rech[3], David Lin[3], Changlu Liu[1], Robyn Miller[2], Jeffrey Schoellerman[1], Jinquan Luo[4], Michael Letavic[3], Bruce Grasberger[2] & Michael Maher [1]

Acid-sensing ion channels (ASICs) are proton-gated cation channels critical for neuronal functions. Studies of ASIC1, a major ASIC isoform and proton sensor, have identified acidic pocket, an extracellular region enriched in acidic residues, as a key participant in channel gating. While binding to this region by the venom peptide psalmotoxin modulates channel gating, molecular and structural mechanisms of ASIC gating modulation by small molecules are poorly understood. Here, combining functional, crystallographic, computational and mutational approaches, we show that two structurally distinct small molecules potently and allosterically inhibit channel activation and desensitization by binding at the acidic pocket and stabilizing the closed state of rat/chicken ASIC1. Our work identifies a previously unidentified binding site, elucidates a molecular mechanism of small molecule modulation of ASIC gating, and demonstrates directly the structural basis of such modulation, providing mechanistic and structural insight into ASIC gating, modulation and therapeutic targeting.

[1] Neuroscience Discovery, Janssen Research & Development, L.L.C., 3210 Merryfield Row, San Diego, CA 92121, USA. [2] Discovery Sciences, Janssen Research & Development, L.L.C., Welsh & McKean Roads, P.O. Box 776, Spring House, PA 19477, USA. [3] Discovery Sciences, Janssen Research & Development, L.L.C., 3210 Merryfield Row, San Diego, CA 92121, USA. [4] Lead Engineering, Janssen Research & Development, L.L.C., Welsh & McKean Roads, P.O. Box 776, Spring House, PA 19477, USA. ✉email: yliu10@its.JNJ.com

Acid-sensing ion channels (ASICs) are proton-gated cation channels widely expressed in nervous systems[1,2]. They are trimeric channels with two-transmembrane domains per subunit and are encoded by at least four genes resulting in homomeric (ASIC1a, ASIC1b, ASIC2a, ASIC3, and ASIC4) and heteromeric channels[3,4]. Mammalian ASIC1a is the most studied ASIC to date, serves as a primary sensor of acidosis in the brain and is involved in a variety of biological processes, including synaptic function and plasticity[5–8], pain sensation[9–11], seizure[12], and neuronal injury[13–17].

ASIC1a can be activated at near neutral pH and can proceed to desensitize from either closed/pre-open or open channels in a pH-dependent manner (closed-state and open-state desensitization, respectively). A multitude of functional studies highlight the critical role of regions in the extracellular domain (ECD), including the acidic pocket (a negative charge-enriched and acidic pair-enriched cavity surrounded by the thumb, finger, and β-ball domains) and the palm domain, in channel activation and desensitization (see ref. [4] for review). Crystal structures of chicken ASIC1 (cASIC1, a homolog of mammalian ASIC1a) in closed[18], open[19,20] and desensitized[21] states provide further molecular and structural insight into the gating mechanism. An emerging picture from these studies indicates a dynamic acidic pocket that adopts an expanded conformation in the closed state due to electrostatic repulsion and contracts in the open and desensitized states with the protonation of the acidic pairs/residues. Furthermore, channel activation induces global conformational changes, including in the palm domain and channel pore/gate, and desensitization results in a conformationally chimeric channel that resembles the closed channel below and open channel above the β11–β12 linkers in the palm domain[18].

Chemical modulators of ASIC1 have been used in functional and structural studies to probe channel gating mechanisms. Venom peptides that target the open[11,19,22], desensitized[22,23] or closed[24] state, for example, have aided in elucidating the molecular underpinnings of channel gating. In particular, psalmotoxin (PcTx1), a prototypical peptidic ASIC1 modulator, is shown to bind at the acidic pocket[19,25] and shift the pH dependence of activation and desensitization by increasing the channel affinity for protons[22].

Amiloride, the most widely used small molecule modulator of ASIC1, blocks the channel by plugging the pore[20,26,27]. As such, it is of limited utility for probing channel gating. Novel, non-amiloride-like small molecule blockers of ASIC channels have been developed for indications such as pain, with no reported molecular mechanism of action[9]. A molecular modeling approach has resulted in a potent, PcTx1-inspired allosteric antagonist thought to bind in the acidic pocket[28]. Other small molecule ligands have been shown to modulate channel gating[29]. However, these studies generally lack direct and substantive demonstration of the molecular/structural basis of gating modulation.

In this study, we report a previously unidentified binding site in ASIC1 for two structurally distinct small molecules, whose occupancy is shown for the first time, to our knowledge, to correlate with and result in modulation of ASIC1 gating. We further demonstrate the molecular mechanism and structural basis of this modulation. Our findings provide important mechanistic and structural insight into modulation of ASIC1 and advance our understanding of the structure, function, and therapeutic targeting of this class of ion channels.

## Results

### JNJ-799760 and JNJ-67869386 are potent allosteric antagonists of ASIC1a.
JNJ-799760 and JNJ-67869386 are small synthetic molecules belonging to two distinct chemical series (Fig. 1a, b).

As shown in Fig. 1c, d, they potently and concentration-dependently inhibit pH 6.8-induced current in a CHO cell stably expressing rat ASIC1a (rASIC1a). The potency and efficacy of this inhibition are both pH dependent (Fig. 1e). At moderate pHs, such as pH 7.1, at which the open probability of ASIC1a is very low (Fig. 1f), both compounds behave like full antagonists with similar potency. At more acidic pHs, such as pH 5.0, antagonism becomes partial with decreased potency. Consistent with antagonism, both compounds decrease the rate of activation and increase the rate of deactivation in a concentration-dependent manner (Supplementary Fig. 1). The functional studies described in this report are performed using rASIC1a.

We studied the effect of JNJ-67869386 on the pH dependence of activation in more detail. As shown in Fig. 1f, JNJ-67869386 shifts the pH dependence of activation towards more acidic values, decreasing both potency and efficacy of proton activation in a concentration-dependent manner. Importantly, the magnitude of both effects reaches a limiting value at saturating concentrations of JNJ-67869386, characteristic of negative modulation via an allosteric mechanism.

The profile of antagonism by JNJ-67869386 and JNJ-799760 differs from that of amiloride, which inhibits ASIC1a noncompetitively by plugging the channel pore. In contrast to JNJ-67869386 and JNJ-799760, amiloride is a full antagonist independent of pH[28]. It also inhibits ASIC1a in a voltage-dependent manner (30 μM amiloride blocks pH 6.0-induced current by $86.8 \pm 1.2\%$ and $62.3 \pm 0.7\%$ at $-80$ and $0\,mV$, respectively; $p < 0.001$, $n = 4$ each; Two-tailed Student's $t$ test; also see ref. [28]), which contrasts with voltage-independent inhibition by JNJ-67869386 (10 nM inhibits pH 6.8-induced currents by $78.2 \pm 5.0\%$ and $75.6 \pm 6.9\%$ at $-80$ and $0\,mV$, respectively; $p > 0.7$, $n = 4$ each; Two-tailed Student's $t$ test). These differences suggest that JNJ-67869386 and JNJ-799760 likely bind to a different site in the channel than amiloride does.

### JNJ-67869386 impedes closed-state desensitization.
Early clues about the effect of JNJ-67869386 on channel desensitization came from intriguing observations of differential current recovery from compound inhibition (wash) at two holding pHs, 7.4 and 8.2. The pH 6.8-evoked current is blocked by 100 nM JNJ-67869386 to near completion at both holding pHs (Fig. 2a, b). Upon switching back to the control solution, the current recovers rapidly (<1 min) and monophasically to the pre-block control level at pH 8.2 (Fig. 2a, c). In contrast, the current recovery at pH 7.4 is bell-shaped: an initial, rapid (<1 min) phase of current increase in which the maximum current amplitude overshoots above the pre-block control level by >50% (wash$_{max}$ in Fig. 2b; Fig. 2c), followed by a slower phase of current decline back to the pre-block control level. These results are summarized over multiple cells in Fig. 2d.

Given the bigger transient pH 7.4 wash current after exposure to JNJ-67869386, we hypothesized that this might be due to a decrease in channel desensitization in the presence of the compound. This would result in more activatable channels at the end of than before compound application, just as observed. This hypothesis is borne out by experiments of the pH dependence of steady-state desensitization. As shown in Fig. 2e, ~60% of the channels are desensitized at pH 7.4 in control buffer. By contrast, 100 nM JNJ-67869386 nearly eliminates all desensitized channels at this pH. More generally, JNJ-67869386 shifts the pH dependence of steady-state desensitization towards acidic pH values (Fig. 2e). Thus, impediment of channel desensitization by JNJ-67869386 presents conditions for an overshoot to occur during current recovery at pH 7.4. Furthermore, the observation that the overshot current eventually declines back to the pre-compound level indicates that the decay phase at pH 7.4 is

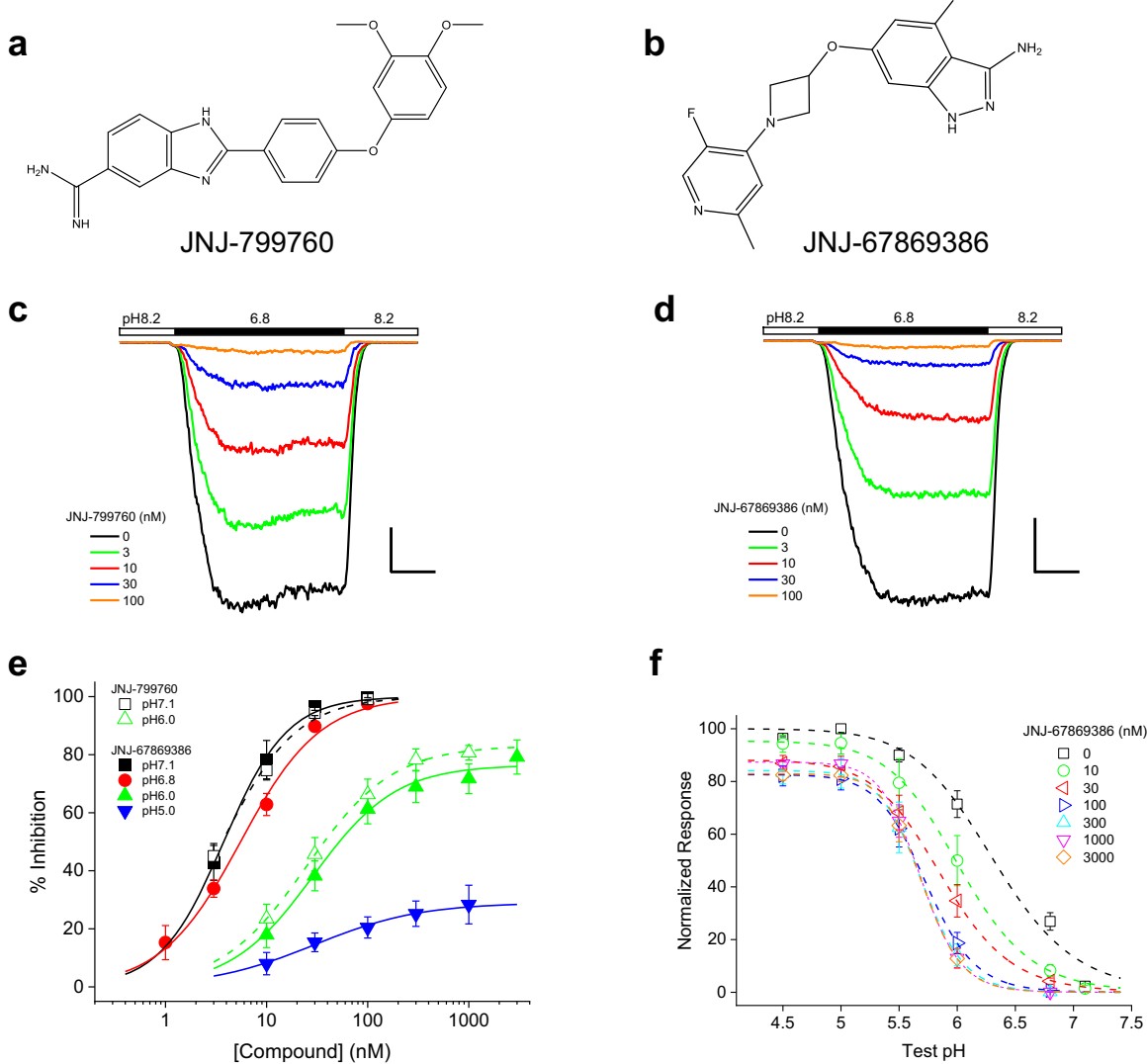

**Fig. 1 JNJ-799760 and JNJ-67869386 are potent and allosteric antagonists of ASIC1a. a** and **b** Chemical structures of JNJ-799760 (**a**) and JNJ-67869386 (**b**). **c** and **d** Traces of pH 6.8-induced current from an ASIC1a-expressing CHO cell showing potent and concentration-dependent inhibition by JNJ-799760 (**c**) and JNJ-67869386 (**d**). Scale bars (horizontal/vertical): 50 ms/100 pA (**c**) and 50 ms/200 pA (**d**), respectively. **e** Concentration–response relationships for current inhibition of ASIC1a by JNJ-67869386 (solid symbols) and JNJ-799760 (open symbols) at various test pHs. For JNJ-67869386, $IC_{50}$ (nM)/max %inhibition $= 3.8 \pm 0.2/100\%$ ($n = 4$), $5.5 \pm 0.4/100\%$ ($n = 6$), $28.9 \pm 3.6/76.5 \pm 2.2\%$ ($n = 7$) and $34.4 \pm 7.6/30.4 \pm 1.8\%$ ($n = 5$) at pH 7.1, 6.8, 6.0 and 5.0, respectively. For JNJ-799760, $IC_{50}$ (nM)/max %inhibition $= 3.6 \pm 0.2/100\%$ ($n = 5$) and $24.9 \pm 1.3/83.1 \pm 1.2\%$ ($n = 4$) at pH 7.1 and 6.0, respectively. **f** pH dependence of channel activation for a range of concentrations of JNJ-67859386. $pH_{50}$/max %response $= 6.31 \pm 0.07/100\%$ ($n = 11$), $6.02 \pm 0.03/95.3 \pm 1.6\%$ ($n = 3$), $5.87 \pm 0.02/88.1 \pm 1.0\%$ ($n = 4$), $5.73 \pm 0.00/82.6 \pm 0.3\%$ ($n = 4$), $5.70 \pm 0.01/84.1 \pm 0.5\%$ ($n = 4$), $5.68 \pm 0.01/87.4 \pm 0.5\%$ ($n = 4$) and $5.70 \pm 0.01/82.9 \pm 0.5\%$ ($n = 4$) for 0, 10, 30, 100, 300, 1000, and 3000 nM JNJ-67859386, respectively. Data at all concentrations except 10 nM are statistically different from control ($p < 0.001$; Two-way ANOVA). Holding pH $= 8.2$ for **c**–**f**.

associated with channel re-desensitization. Consistent with this scenario, development of desensitization at pH 7.4 follows a time course resembling the kinetics of the decay phase in the wash experiments (both >1 min; Supplementary Fig. 2).

The bell-shaped current recovery also suggests that the rate of current recovery from compound inhibition (associated with the compound dissociation rate) is faster than that of channel desensitization. Consistent with this, recovery from JNJ-67869386 inhibition has a time constant of ~13 s, at least at both pH 8.2 and pH 7.8 (Supplementary Fig. 2). Assuming similar recovery kinetics at pH 7.4 (which is not unreasonable given the apparent lack of pH dependence between pH 8.2 and pH 7.8), the onset of desensitization, which is >1 min (Supplementary Fig. 2), would indeed be too slow to prevent recovered current from overshooting.

Figure 2e also explains the pH 8.2 results in Fig. 2c, d. Since channels are not desensitized at pH 8.2, no more channels are recoverable from desensitization after exposure to JNJ-67869386. Thus, no overshoot occurs at this pH and recovery from JNJ-67869386 inhibition simply follows a monophasic time course that just reflects compound dissociation from closed channels.

**Effects of JNJ-67869386 and J-799760 on the kinetics of closed-state desensitization**. We have shown above that JNJ-67869386 impedes steady-state desensitization. To understand the mechanism by which this occurs, we next studied the effects of JNJ-67869386 on the kinetics of (i.e., rates of development of and recovery from) closed-channel desensitization. As shown in Fig. 3a, c, pH 7.1-induced desensitization develops with a time constant of 3.4 s in the absence of JNJ-67869386. JNJ-67869386

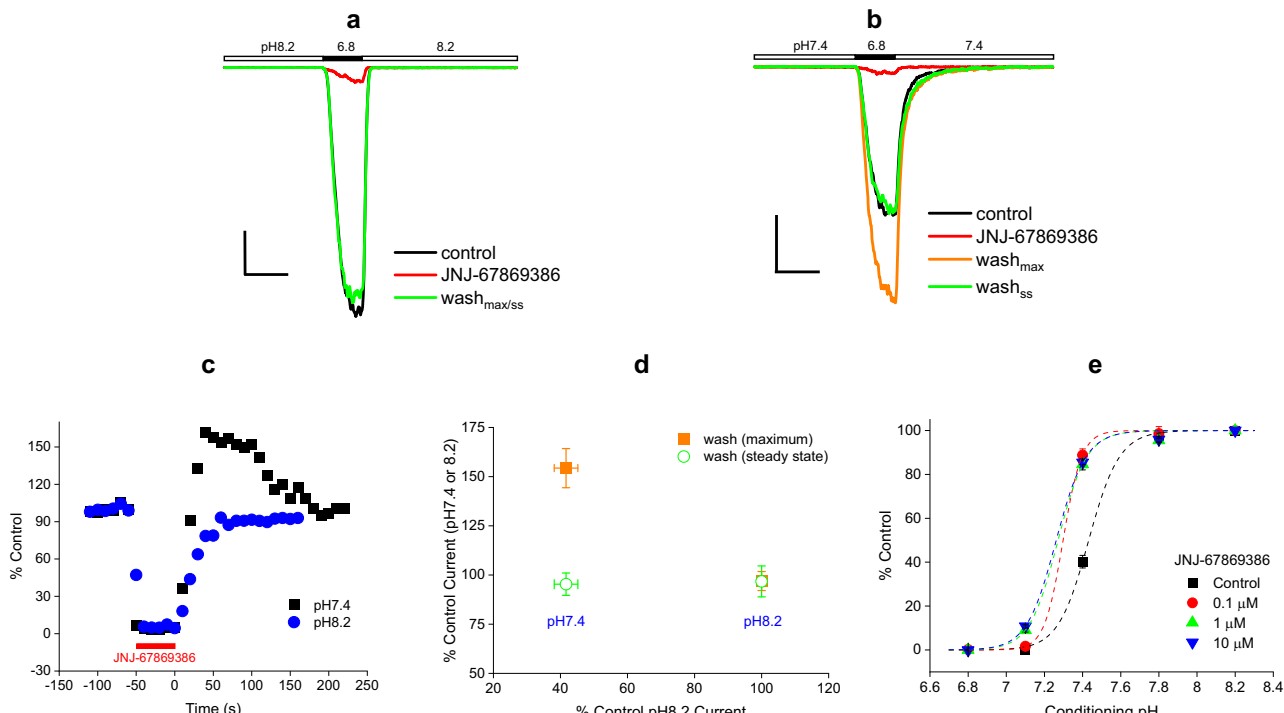

**Fig. 2 Effect of JNJ-67869386 on the pH dependence of recovery from inhibition and of steady-state desensitization. a** and **b** Traces of pH 6.8-induced currents from an ASIC1a-expressing cell with and without 100 nM JNJ-67869386 at the holding pH of 8.2 (**a**) and 7.4 (**b**). Scale bars: 100 ms/100 pA (horizontal/vertical). **c** Time courses of inhibition of pH 6.8-induced current by 100 nM JNJ-67869386 and recovery (wash) from inhibition at the holding pH of 8.2 and 7.4 from the same cell. Current amplitudes are normalized to that of pre-JNJ-67869386 controls at the respective pHs. Wash starts at $t = 0$ s. **d** Summary of all experiments like (**c**). The average normalized maximal (solid squares) and steady-state (open circles) wash current amplitude ($n = 4$ for both pH 7.4 and 8.2) is plotted against the pre-JNJ-67869386 control current amplitude (normalized to the pH 8.2 control; $n = 4$). The maximal and steady-state wash values at pH 7.4 are statistically different ($p < 0.01$; Two-tailed Student's $t$-test). $V_h = -60$ mV. **e** Effect of JNJ-67869386 (0.1–10 μM) on the pH dependence of steady-state desensitization. $pH_{50} = 7.43 \pm 0.01$ ($n = 14$), $7.30 \pm 0.03$ ($n = 8$), $7.27 \pm 0.01$ ($n = 4$) and $7.26 \pm 0.01$ ($n = 4$) for control, 0.1, 1, and 10 μM JNJ-67869386, respectively. Data at all concentrations are statistically different from control ($p < 0.001$; Two-way ANOVA).

(100 nM) slows down this process considerably, to 15.1 s (Fig. 3b, c), consistent with the results in Fig. 2e showing impediment to steady-state desensitization. Recovery (at pH 8.2) from pH 7.1-induced desensitization in the absence of JNJ-67869386 is rapid and can be fitted to a single exponential function with a time constant of 231 ms (Fig. 3d, f). Intriguingly, the time course of recovery in the presence of 100 nM JNJ-67869386 is bell-shaped and can be fitted to a double exponential function (Fig. 3e, f). Here, the current amplitude first rises rapidly to a peak value well above (>200%) the pre-desensitization level, with kinetics resembling that of recovery in the absence of JNJ-67869386 ($\tau = 381$ ms). It then declines more slowly back to the pre-desensitization level with a time constant of 1.4 s, resembling the slower kinetics of current inhibition by JNJ-67869386 (Supplementary Fig. 3). Since JNJ-67869386 is present throughout the experiment, the overshoot and bell-shaped kinetics are both readily explicable if JNJ-67869386 is assumed to dissociate from the channel during the pH 7.1 pulse. In this scenario, more channels are unblocked at the end than at the beginning of channel desensitization. These (unblocked) channels recover from desensitization as quickly as in control buffer (because compound inhibition is too slow to catch up during this phase), resulting in an overshoot. Recovered channels subsequently undergo (slower) re-inhibition in the continued presence of JNJ-67869386, completing the biphasic time course. (Note: the overshoot is much more pronounced using pH 6.0 as test pulse than using pH 5.0, due to the much higher efficacy of JNJ-67869386 at pH 6.0.)

JNJ-799760 (100 nM) exhibits a similar biphasic time course of recovery from desensitization and re-inhibition (Fig. 3f), again suggesting a shared mechanism with JNJ-67869386. (The smaller overshoot and faster decay kinetics for JNJ-799760 are consistent with its faster rate of current inhibition as shown in Supplementary Fig. 3.) By comparison, the kinetics of recovery from pH 7.1-induced desensitization in the presence of amiloride (30 μM) are not different from that in control buffer (Fig. 3f), consistent with amiloride having a different mechanism than JNJ-67869386 and JNJ-799760.

**Effects of JNJ-67869386 and J-799760 on the kinetics of open-state desensitization**. In contrast to closed-state desensitization, pH 5.0-induced open-channel desensitization is accelerated by JNJ-67869386 and JNJ-799760 (Fig. 4a, b), suggesting that these compounds may destabilize the open state. Because open-channel desensitization appears more (rather than less, as in closed-state desensitization) favored (at least relative to the open state) in the presence of these compounds than in their absence, we wondered whether open-channel and closed-channel desensitization might affect compound binding differently. To this end, we examined the effect of JNJ-67869386 on recovery from pH 5.0-induced desensitization. As shown in Fig. 4c, e, the time course of recovery at pH 8.2 in the absence of JNJ-67869386 is the same as that from closed-state desensitization ($\tau = 250.2$ ms), as would be expected if recovery at pH 8.2 from both closed-channel and open-channel desensitization occurs via the same pathway[30]. (Note that

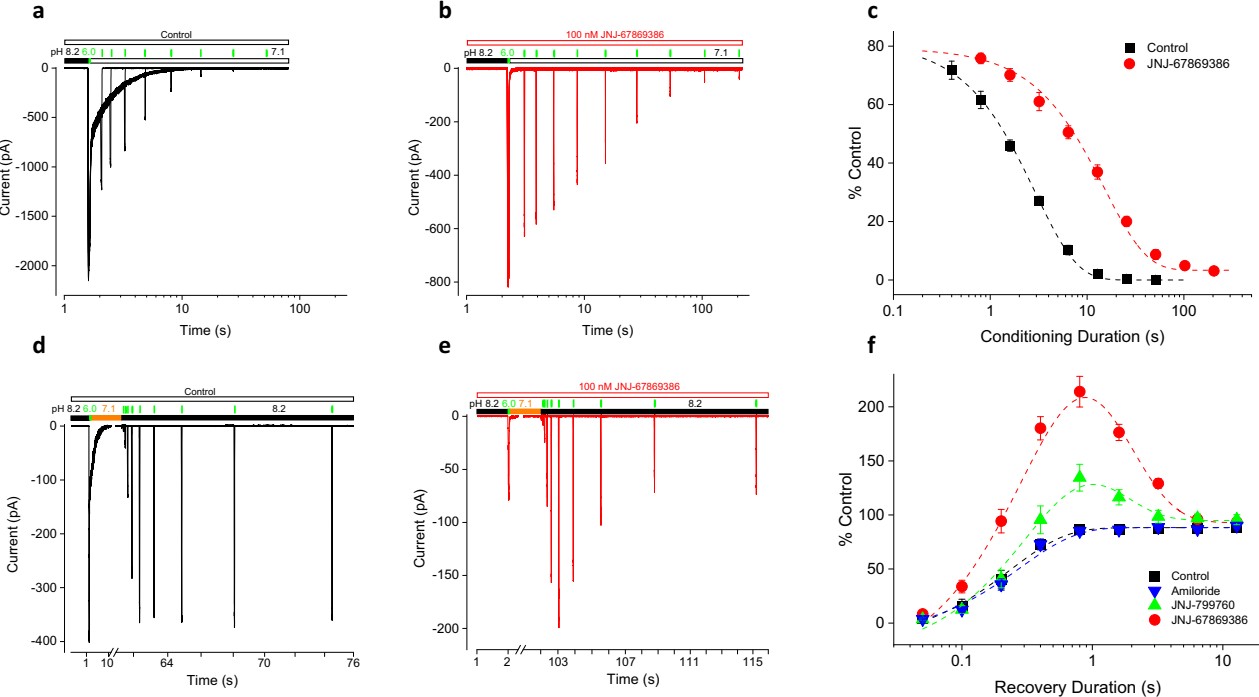

**Fig. 3 Effect of JNJ-67869386 and JNJ-799760 on the kinetics of closed-state desensitization. a** and **b** Current traces showing the time course of development of pH 7.1-induced desensitization without (**a**) or with (**b**) 100 nM JNJ-67869386. Brief pH 6.0 test pulses (thin vertical bars above the current traces) are applied immediately before and at various times during the pH 7.1 application. Inter-pulse interval = 120 s (**a**) and 150 s (**b**). **c** Summary of all experiments like (**a**) and (**b**). Data are normalized to the first pH 6.0 (pre-pH 7.1) peak and fitted to a single exponential function with time constants of 2.9 ± 0.1 s (control; $n = 6$) and 15.1 ± 1.3 s (JNJ-67869386; $n = 6$), respectively. The JNJ-67869386 data are statistically different from control ($p < 0.001$; Two-way ANOVA). **d** and **e** Current traces showing the time course of recovery from pH 7.1-induced desensitization without (**d**) or with (**e**) 100 nM JNJ-67869386. A 60/100 s (control/JNJ-67869386) pH 7.1 pulse is preceded by a brief pH 6.0 pulse and followed by a pH 8.2 pulse interspersed with brief pH 6.0 pulses (thin vertical bars above the current traces) at various intervals. **f** Summary of all experiments like (**d**) and (**e**). Data are normalized to the first pH 6.0 (pre-pH 7.1) peak and fitted to a single (control/30 μM amiloride) or double (100 nM JNJ-67869386/JNJ-799760) exponential function with time constants of 231.5 ± 17.4 ms (control; $n = 6$), 277.6 ± 35.9 ms (amiloride; $n = 4$), 381.0 ± 85.1 ms/1.4 ± 0.4 s (JNJ-67869386; $n = 6$) and 400.0 ms/1.0 ± 0.4 s (JNJ-799760; $n = 6$), respectively. Data for JNJ-67869386 and JNJ-799760 ($p < 0.001$ for both), but not amiloride ($p > 0.5$), are statistically different from control (two-way ANOVA).

recovery from pH 5.0-induced desensitization is significantly less complete than that from pH 7.1-induced desensitization due to the entrance of some channels into a long-lived desensitized state). In the presence of JNJ-67869386, the rate of recovery from pH 5.0-induced desensitization also exhibits a biphasic time course with virtually the same time constants as those for pH 7.1-induced desensitization (Fig. 4d, e). These results suggest that, as with closed-state desensitization, compound dissociation also occurs during pH 5.0-induced open-state desensitization.

**JNJ-67869386 and JNJ-799760 do not bind to desensitized channels**. Taken together, data from Figs. 3 and 4 are consistent with a scenario in which JNJ-67869386 and JNJ-799760 bind to closed but dissociate from desensitized channels. As channels recover from desensitization (at pH 8.2 and in the presence of compound), compound rebinding to recovered (i.e., closed) channels causes re-inhibition. We tested this idea further by preincubating compound with initially closed (at pH 8.2) and subsequently desensitized (at pH 7.1 for 100 s) channels but by examining recovery from desensitization (at pH 8.2) in the absence of compound. As shown in Fig. 5a, most of the channels here recover quickly with a time constant resembling that of recovery from desensitization in the absence of compound. This suggests that despite being bound to compound at the start of and exposed to compound throughout the desensitization period, most channels start recovery from desensitization with no

compound bound, consistent with the conclusions reached from Fig. 3. A small fraction of channels still recovers slowly, resembling the kinetics of current recovery from compound inhibition in the closed state (Fig. 5a), suggesting that some channels are still compound bound at the start of recovery and that the desensitizing pulse is not long enough for complete compound dissociation to occur.

If complete compound dissociation occurs at equilibrium in the desensitized state, then the compound would not be expected to bind to channels that are already desensitized. To test this hypothesis, we performed experiments in which compound was only present during the pH 7.1 desensitizing pulse. With this protocol, most channels are desensitized before compound binding can occur. As shown in Fig. 5b, the recovery time courses are similarly fast for control and compound and are monophasic without a slow component, indicating that JNJ-67869386 and JNJ-799760 indeed do not bind to pre-desensitized channels. Together with Fig. 3, these results indicate that, at equilibrium, channels desensitized via the closed state are not compound bound.

We also examined the kinetics of recovery from open-state desensitization using a protocol similar to that in Fig. 5a for closed-state desensitization (except for the pH and duration of the desensitizing pulse). Since pH 5.0-induced desensitization occurs faster in the presence of JNJ-67869386 or JNJ-799760, the compound is necessarily bound during channel opening and only dissociates significantly from the channel subsequently over the

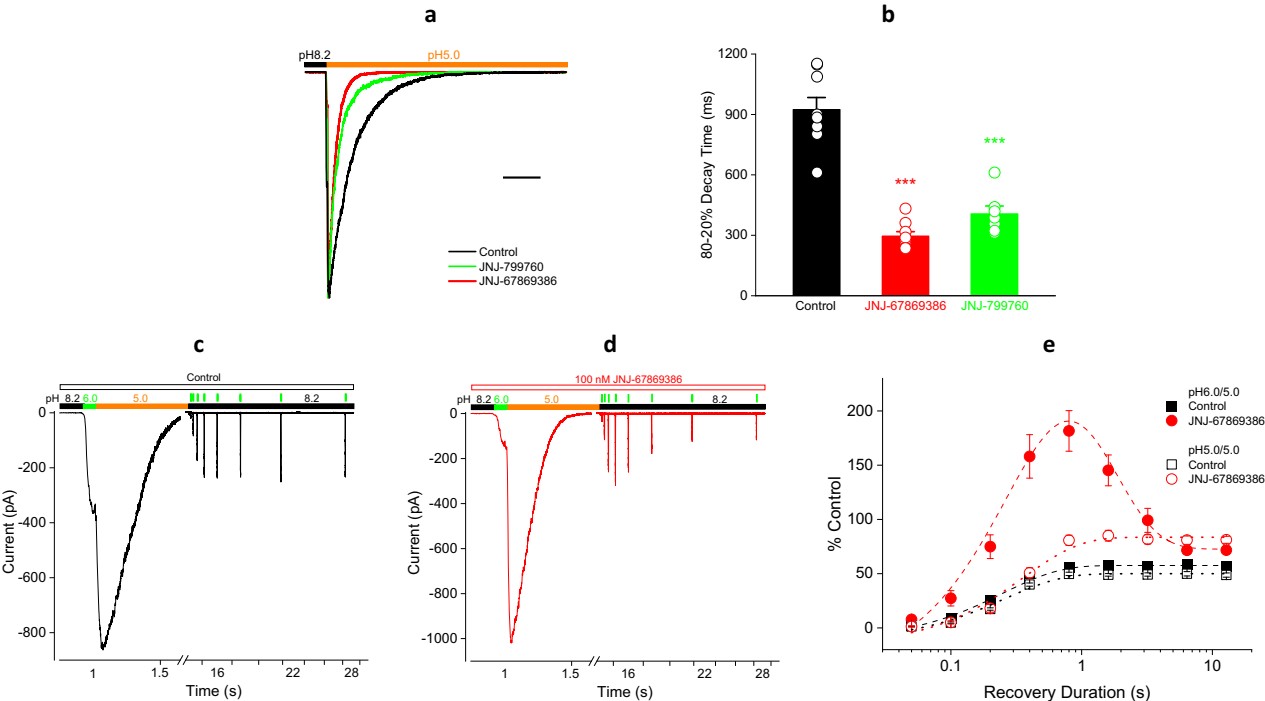

**Fig. 4 Effect of JNJ-67869386 and JNJ-799760 on the kinetics of open-state desensitization. a** Current traces (scaled to the control peak) showing pH 5.0-induced desensitization in control, JNJ-67869386 and JNJ-799760 (both 100 nM). Scale bar: 1 s. **b** Summary of 80–20% current decay time for all experiments like (**a**). Data for JNJ-67869386 ($n = 10$) and JNJ-799760 ($n = 7$) are both statistically different from control ($n = 9$) ($p < 0.001$; one-way ANOVA). Data for individual cells are shown in open circles. **c** and **d** Current traces showing the time course of recovery from pH 5.0-induced desensitization without (**c**) and with (**d**) 100 nM JNJ-67869386. A 14 s pH 5.0 pulse is immediately preceded by a brief pH 6.0 test pulse and followed by a pH 8.2 pulse interspersed with brief pH 6.0 test pulses at various intervals (thin vertical bars above current trace). In some experiments, brief pH 6.0 pulses are replaced with brief pH 5.0 pulses. **e** Summary of all experiments like (**c**) and (**d**). Data are normalized to the first pH 6.0 (solid symbols) or pH 5.0 (open symbols) peak and fitted to a single or double exponential function with time constants of 250.2 ± 17.3 ms (test/desensitizing pH = pH 6.0/5.0 control; $n = 9$), 369.8 ± 113.8 ms/1.2 ± 0.5 s (pH 6.0/5.0 JNJ-67869386; $n = 5$), 258.6 ± 36.6 ms (pH 5.0/5.0 control; $n = 6$) and 358.6 ± 56.2 ms (pH 5.0/5.0 JNJ-67869386; $n = 7$), respectively. The JNJ-67869386 data for both pH 6.0/5.0 and pH 5.0/5.0 are statistically different from the respective control ($p < 0.001$; two-way ANOVA).

course of desensitization. Indeed, when the pH 5.0 desensitizing pulse is relatively short (14 s), the kinetics of recovery from desensitization for JNJ-67869386 is biphasic with the fast and slow components resembling those of recovery from the compound-unbound desensitized state and compound dissociation from the closed state, respectively (Fig. 5c), similar to that shown in Fig. 5a. This indicates that some desensitized channels are still compound bound after 14 s. When the duration of the pH 5.0 pulse is increased to 42 s, however, only the fast component remains. The absence of the slow component indicates that after 42 s in pH 5.0, channels are no longer compound bound. For JNJ-799760, 14 s in pH 5.0 is sufficiently long for eliminating the slow component (Fig. 5c), consistent with its faster rate of dissociation than JNJ-67869386. Thus, as with closed-state desensitization, compound unbinds from channels desensitized via the open state as well.

Lastly, we investigated whether compound binding occurs to pre-desensitized (via the open state by pH 5.0) channels by applying compound only during the pH 5.0 pulse (14 s). Here, open channels are not compound bound because the rate of desensitization is much faster than that of compound association. This is also supported by the observation that there is no difference in the rate of pH 5.0-induced desensitization between control and compound (Supplementary Fig. 4). Under these conditions, the rate of recovery from desensitization is indistinguishable between control and compound applications (Fig. 5d), confirming that, as with closed-state desensitization,

these compounds cannot bind to channels pre-desensitized via the open state either.

**Kinetic model.** Our findings thus far can be qualitatively appreciated with a simplified kinetic scheme shown in Fig. 6a. In the closed state (C), high-affinity compound binding drives channels to the closed/blocked state (CB). Closed-state desensitization (e.g., at pH 7.1) proceeds from C → CD and from CB → C → CD in the absence and presence of compound, respectively, with the latter being slower due to the extra step of compound unbinding before desensitization. CBD is a transient passthrough from OBD (primarily) to CB; transitions into CBD do not otherwise take place. Activation of compound-bound, closed channels (CB → OB) is shifted towards more acidic pHs. Compound-bound open channels (OB) are less stable and conductive than unbound open channels (O), and enter a transient, desensitized state (OBD) from which they transition to a stable, compound-unbound desensitized state (OD).

Results from kinetic modeling further support the conclusion that compound-bound desensitized states (e.g., CBD in Fig. 6a) do not exist at equilibrium. To test whether compound-bound desensitized states are necessary to account for the experimental data, we performed kinetic simulations using the linear scheme in Fig. 6b, which explicitly excludes compound-bound desensitized states. Reflecting the trimeric nature of ASIC1a, the scheme in Fig. 6b contains seven states: a (fully) closed state (C) and a state each with one, two or three desensitized or compound-bound

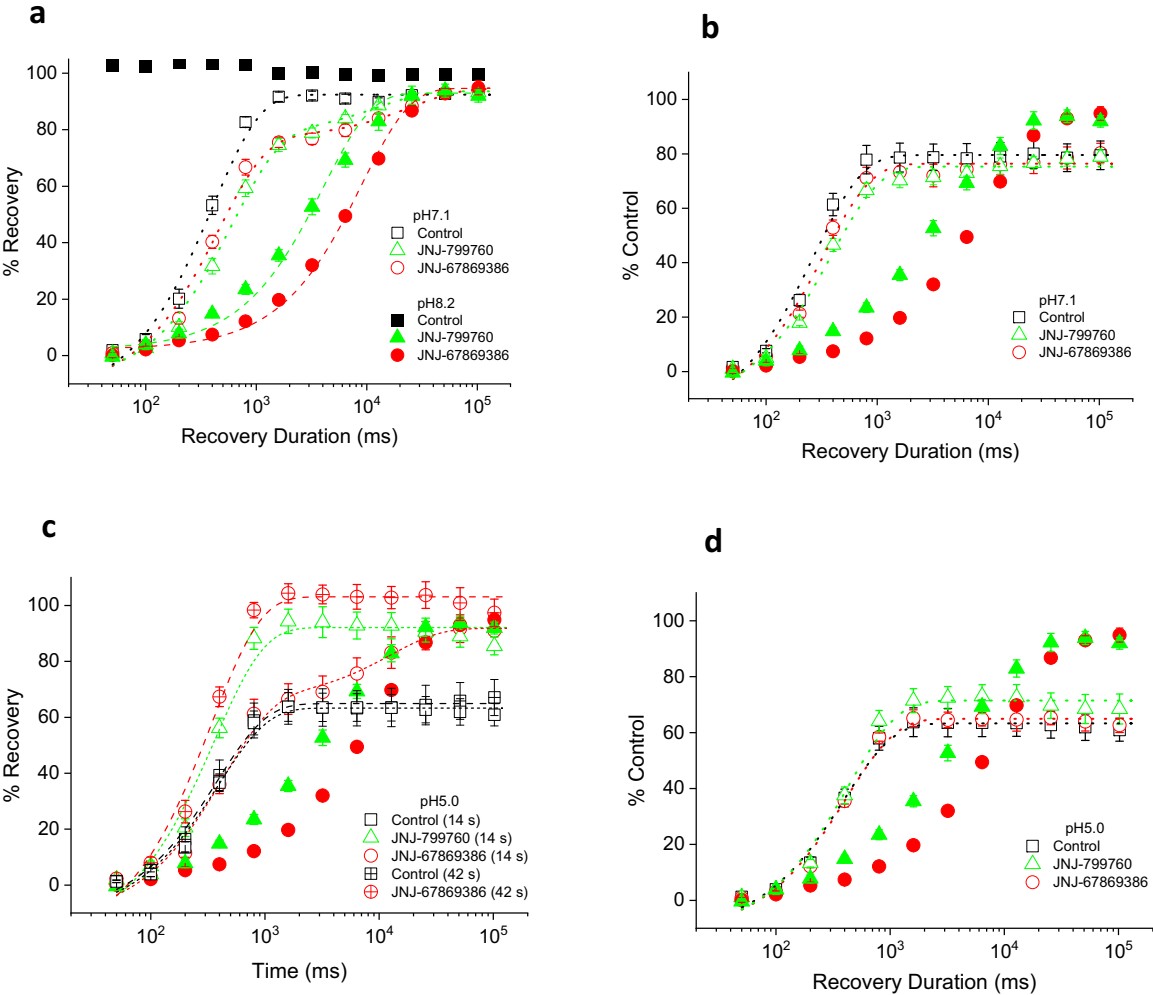

**Fig. 5 Effect of JNJ-799760 and JNJ-67869386 on the kinetics of compound-free recovery from desensitization. a** Recovery from pH 7.1-induced-desensitization (open symbols) or from inhibition at pH 8.2 (solid circles and triangles) by JNJ-67869386 and JNJ-799760 (both 100 nM). Compound (except pH 8.2 control, solid squares) is present only prior to recovery. Recovery $\tau = 402.2 \pm 34.7$ ms, $450.5 \pm 57.5$ ms/$25.1 \pm 17.9$ s (18.8%), $598.0 \pm 75.0$ ms/$12.3 \pm 8.3$ s (16.9%), $9.0 \pm 0.5$ s and $4.0 \pm 0.3$ s, respectively, for pH 7.1 ($n = 5$, same cells) control, JNJ-67869386, JNJ-799760, and pH 8.2 ($n = 11$) JNJ-67869386 and JNJ-799760. The pH 7.1 JNJ-67869386 and JNJ-799760 data are statistically different from pH 7.1 control and the respective pH 8.2 data ($p < 0.001$). **b** Compound (100 nM) only during pH 7.1 ($n = 5$, same cells). $\tau = 280.3 \pm 45.8$ ms (control), $310.4 \pm 52.0$ ms (JNJ-67869386; $p < 0.05$), and $357.5 \pm 53.0$ ms (JNJ-799760; $p < 0.001$), respectively. **c** Recovery from pH 5.0-induced desensitization with compound only before recovery. $\tau = 379.3 \pm 31.3$ ms (control$_{42}$, 42 s; $n = 4$), $339.4 \pm 36.5$ ms (JNJ-67869386, 42 s; $n = 4$; $p < 0.001$ vs. both control$_{42}$ and pH 8.2), $390.8 \pm 39.4$ ms (control$_{14}$, 14 s; $n = 9$; $p > 0.3$ vs. control$_{42}$), $425.1 \pm 74.0$ ms/$12.6 \pm 6.1$ s (28.3%) (JNJ-67869386, 14 s; $n = 10$; $p < 0.001$ vs. both control$_{14}$ and pH 8.2) and $360.4 \pm 46.4$ ms (JNJ-799760, 14 s; $n = 7$; $p < 0.001$ vs. both control$_{14}$ and pH 8.2), respectively. **d** Compound (100 nM) only during pH 5.0. $\tau = 390.8 \pm 39.4$ ms (control; $n = 9$), $414.0 \pm 42.2$ ms (JNJ-67869386; $n = 4$; $p > 0.3$) and $426.5 \pm 47.7$ ms (JNJ-799760; $n = 4$; $p < 0.01$), respectively. Holding pH = 8.2. Data are normalized to the control peak before compound and fitted to a single/double exponential function. Steady-state current is subtracted before normalization. Same pH 8.2 data (solid symbols) in **a**–**d**. Statistical tests are two-way ANOVA.

subunits (CD$_1$, CD$_2$, CD$_3$ and CB$_1$, CB$_2$, CB$_3$), respectively (an expanded version of the three states, C, CD, and CB, in Fig. 6a). With experimentally based assumptions and values for the parameters in the model (see the "Methods" section), our simulations produce broadly good fits to the experimentally observed kinetic data, including the onset of (Fig. 6c) and recovery from (Fig. 6d) closed-state desensitization as well as kinetics of compound block and unblock (Fig. 6e), demonstrating that compound-bound desensitized states are unnecessary. Furthermore, the same set of parameter values also produce simulated responses that match the kinetics of recovery from open-state desensitization (Fig. 6f), indicating that recovery from equilibrated open-state desensitization also originates from OD rather than OBD. Thus, results from quantitative kinetic

modeling strongly support the contention that desensitized channels are not compound bound at equilibrium.

**JNJ-67869386 impedes tachyphylaxis and the effect of PcTx1.** Repeated stimulations of ASIC1a, particularly at low pHs, cause tachyphylaxis, a process that depends on permeation of both H$^+$ and Ca$^{2+}$ ions[31] and whereby channels enter a long-lived, desensitized state. As shown in Fig. 7a, tachyphylaxis is more pronounced at pH 5.0 than at pH 7.1, consistent with literature findings and with our observation that recovery from pH 7.1-induced desensitization (Fig. 3f) is more complete than that from pH 5.0-induced desensitization (Fig. 4e). JNJ-67869386 (100 nM) significantly decreases tachyphylaxis at both pHs (Fig. 7a),

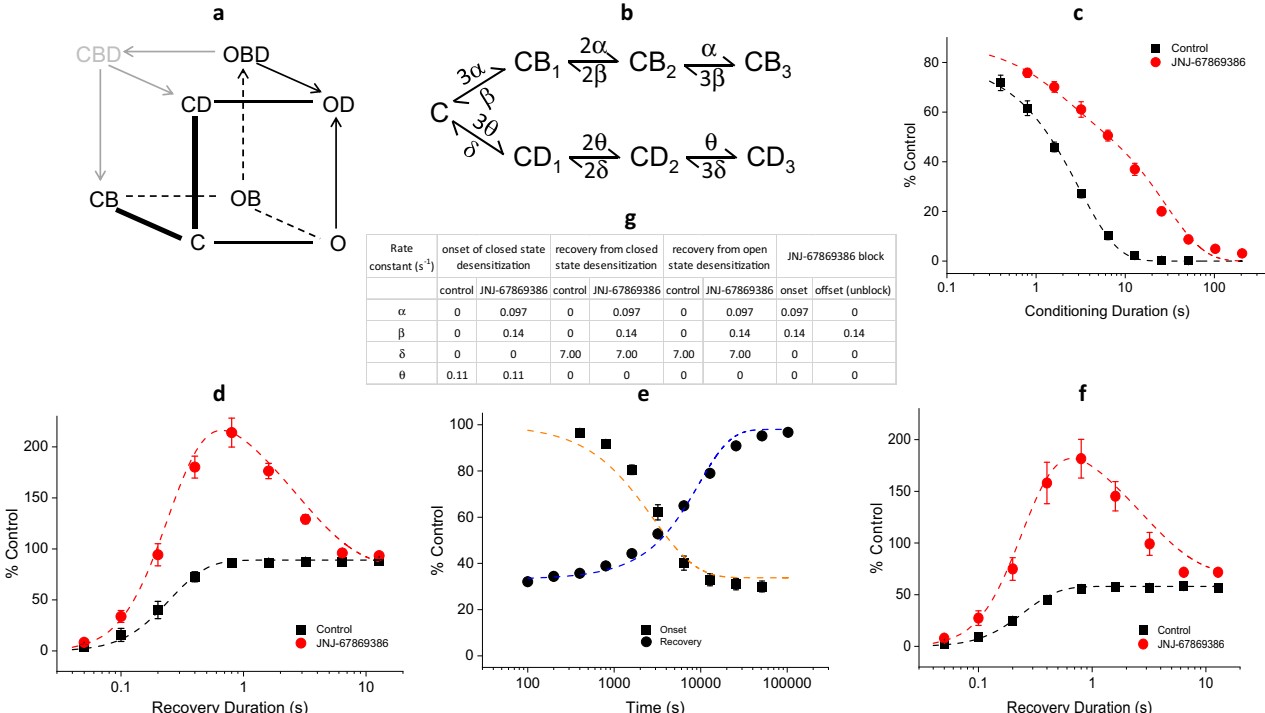

**Fig. 6 Kinetic modeling of the effect of JNJ-67869386 on desensitization kinetics. a** Qualitative model for channel activation and desensitization in the absence and presence of compound. In the presence of compound, closed-state desensitization and recovery involve only three states, CB, C, and CD (transitions between these states are represented by thick lines). Recovery from (equilibrated) open-state desensitization funnels through the same three states, after an initial rapid transition from OD → CD. Recovery from (unequilibrated) open-state desensitization, i.e., before compound dissociation (OBD → OD) is completed, takes an additional path via CBD (grayed), a pass-through state, as follows: OBD → CBD → CB → C. Arrowed lines indicate irreversible transitions. **b** Linear kinetic model used for kinetic simulations. Compound-bound desensitized states are excluded from the scheme. **c** Simulated responses (dashed lines) of the onset of closed-state desensitization overlaid with experimental data from Fig. 3c. **d** Simulated responses (dashed lines) of recovery from closed-state desensitization overlaid with experimental data from Fig. 3f. **e** Simulated responses (dashed lines) of onset of and recovery from compound inhibition overlaid with experimental data from Fig. 5a (without steady-state subtraction) and Supplementary Fig. 3. **f** Simulated responses (dashed lines) of recovery from open-state desensitization overlaid with experimental data from Fig. 4e. **g** Rate constants used for the simulations. All simulations are performed with the same set of parameter values given in the table (taking into account irreversible transitions as appropriate). For recovery from desensitization, the initial state is set to be $CD_3$.

consistent with the observation that recovery from pH 5.0-induced desensitization is more complete in the presence of JNJ-67869386 than that in control buffer (Fig. 4e). As such, JNJ-67869386 inhibits both fast (short-lived) and slow (long-lived) desensitized states.

In contrast to JNJ-67869386, PcTx1 promotes desensitization[22]. We studied how PcTx1 and JNJ-67869386 might interact to modulate ASIC1a. As shown in Fig. 7b, PcTx1 (10 nM) rapidly inhibits ASIC1a current, which, not surprisingly, is significantly hindered by JNJ-67869386 (100 nM). Kinetics of recovery from PcTx1 inhibition in the absence and presence of JNJ-67869386 are similar (Fig. 7c), suggesting that PcTx1 and JNJ-67869386 binding may be mutually exclusive. Consistent with the kinetic effect, JNJ-67869386 significantly reverses the PcTx1-induced pH shifts in steady-state desensitization (Fig. 7d).

**JNJ-799760 binds to a site at the acidic pocket in the closed state of the channel**. To understand the structural basis underlying the ASIC1a modulation by these molecules, we obtained the X-ray structure of a construct of cASIC1 containing residues 26–463 of the full-length polypeptide (ΔASIC1) in complex with JNJ-799760 (ΔASIC1/JNJ-799760; PDB code: 6X9H). Chicken ASIC1 and rat ASIC1a are ~90% identical and have similar pH sensitivity and pharmacology, making it likely that these channels have generally conserved structural conformations. The ΔASIC1

construct we used is identical to that published by Jasti et al.[3] (PDB code: 2QTS; pH 6.5) and Dawson et al. [25] (PDB code: 3S3W; pH 7.5). It was described as expressing a non-functional channel[3], although we observed significant pH-sensitive currents at holding pH 8.2 in ΔASIC1-transfected (but not untransfected) cells.

The crystal of the ΔASIC1/JNJ-799760 complex is grown at pH 7.5 and diffracted to 3.0 Å resolution (Table 1). It belongs to the $P2_12_12_1$ space group and contains a ΔASIC1 trimer and three JNJ-799760 molecules (one per subunit chain) in the asymmetric unit. The three chains share similar overall structural conformations: the root mean square displacement (RMSD) values for superposing chains A & B, A & C, and B & C are 0.336, 0.331, 0.357, respectively. The overall architecture of the JNJ-799760-bound trimeric channel resembles that described in the literature[18], with each subunit taking the shape of a clenched fist composed of a palm, wrist, finger, knuckle, thumb, β-ball domain, and two TM domains.

The JNJ-799760-binding site maps to the finger and thumb domains within each subunit (Fig. 8a, b), at the edge of the acidic pocket and surrounded by α3, α5, portions of β2–α1, α1–α2 and α4–α5 linkers, and the acidic loop before β6 (Fig. 8c). JNJ-799760 makes extensive hydrophobic interactions with L115, L116, M155, L231, P232, M326, V327, and Y341, forms π–π interactions with the aromatic rings of F99 and Y159, as well as hydrogen bonds with the carboxyl side chains of E98, the

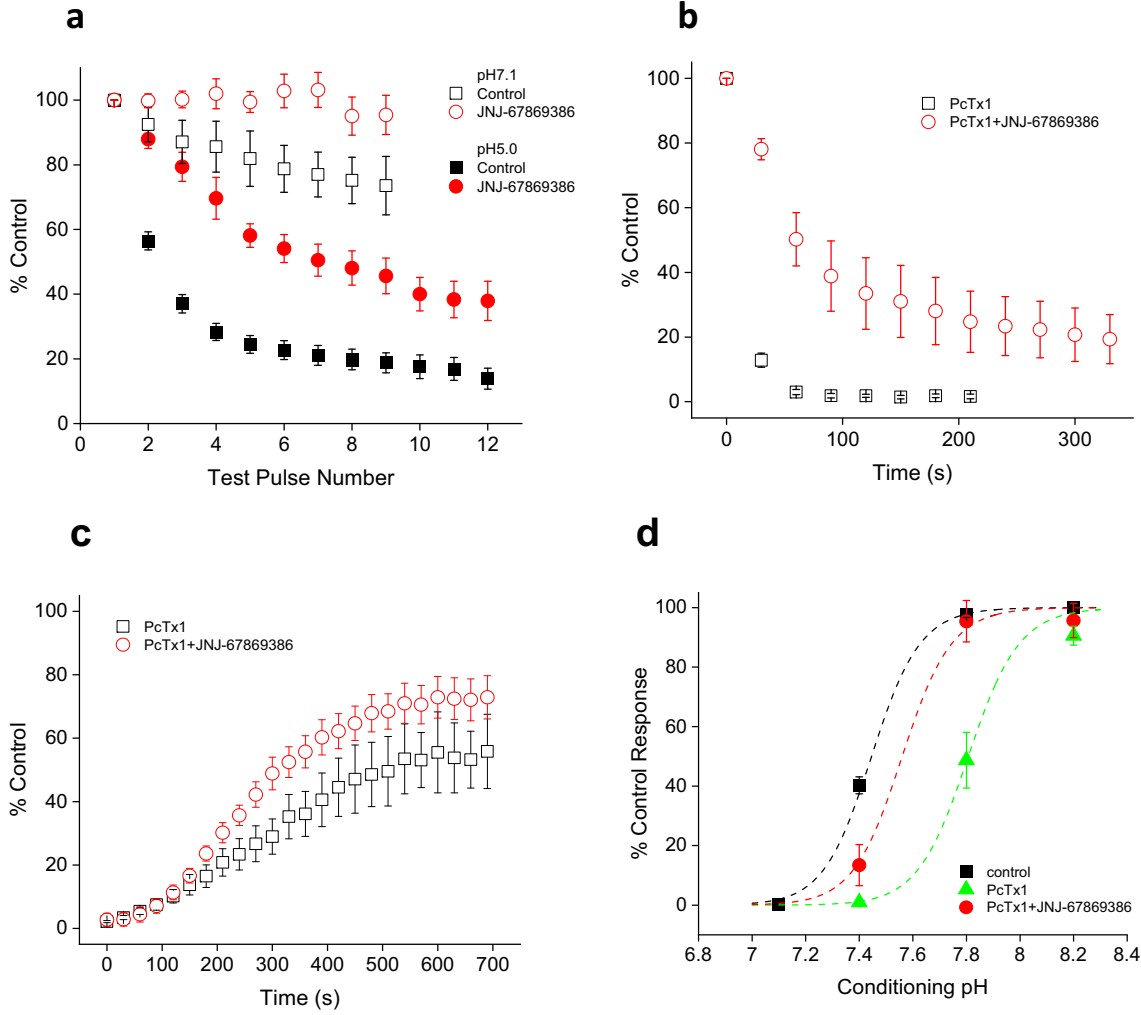

**Fig. 7 Effect of JNJ-67869386 on tachyphylaxis and PcTx1-induced inhibition. a** Conditioning pH 7.1-induced (open symbols) or pH 5.0-induced (solid symbols) induced tachyphylaxis in the absence (squares) or presence (in all pH buffers; circles) of 100 nM JNJ-67869386. Conditioning durations/Inter-pulse intervals are 60/120 s (pH 7.1 control; $n = 7$), 100/150 s (pH 7.1 JNJ-67869386; $n = 6$) and 14/60 s (pH 5.0 control and JNJ-67869386; $n = 12$ each). Data for both pH 7.1 and pH 5.0 are statistically different between control and JNJ-67869386 ($p < 0.001$). Data are normalized to the peak of pre-conditioning test pulse. Holding pH = 8.2. **b** Kinetics of current inhibition by PcTx1 (10 nM) in the absence (squares; $n = 6$) and presence (in all pH buffers; circles; $n = 5$; $p < 0.001$) of 100 nM JNJ-67869386. Holding pH = 7.4. **c** Kinetics of current recovery (wash) from PcTx1 (10 nM) inhibition in the absence (squares; $n = 4$) and presence (in all pH buffers; circles; $n = 4$; $p < 0.001$) of 100 nM JNJ-67869386. PcTx1 is removed at $t = 0$ s. Holding pH = 7.4. **d** pH dependence of steady-state desensitization in control (squares), 10 nM PcTx1 (triangles) or 10 nM PcTx1 + 100 nM JNJ-67869386 (circles; JNJ-67869386 in all pH buffers). Data are fitted to a logistic function (dashed lines) with $pH_{50}$ values of 7.43 ± 0.01 (control; $n = 14$), 7.81 ± 0.02 (PcTx1; $n = 4$; $p < 0.001$ vs. control) and 7.56 ± 0.03 (PcTx1+JNJ-6786386; $n = 4$; $p < 0.001$ vs. PcTx1 and $p < 0.01$ vs. control), respectively. All statistical tests are two-way ANOVA.

backbone carbonyls of L116, L231, and P232, and the backbone nitrogen of V327 (Fig. 8d). There is a small difference for chain B where only one hydrogen bond is formed with the carbonyl of P232 (instead of two for chains A and C).

To determine the state of the channel in ΔASIC1/JNJ-799760, we first determined a set of key structural conformations in several regions of the channel for ΔASIC1/JNJ-799760 vis-à-vis 3S3W[25], the apo structure obtained at the same pH as ΔASIC1/JNJ-799760 (pH 7.5). We then determined the state of the channel by comparing these conformations in ΔASIC1/JNJ-799760 with those of published structures (grouped in Supplementary Table 1 based on channel states).

Relative to 3S3W, the ΔASIC1/JNJ-799760 channel displays significant and global conformational differences, as shown in Supplementary Table 1. First, the acidic pocket of ΔASIC1/JNJ-799760 adopts an expanded conformation, as evidenced by the

outward pivots/shifts of the thumb helices α4 (up to 2.6 Å) and α5 (up to 5.3 Å) away from the channel core and the acidic loop (Fig. 9a) and by the increased distances between the side chains of acidic pairs, D238–D350 and E239–D346, in the acidic pocket (Fig. 9b). Significantly, JNJ-799760 disrupts the helical integrity of α5, a continuous helix in 3S3W (and other ASIC1 crystal structures published to date), breaking it into two separate halves, α5a and α5b (Figs. 8a and 9a). Second, the orientation of the T84–R85 peptide bond in the β1–β2 linker undergoes a ~180° flip in ΔASIC1/JNJ-799760 relative to that in 3S3W and other desensitized structures[3,21,32] (Fig. 9c), coming in line with the orientation in closed-channel structures (Supplementary Table 1 and Supplementary Fig. 5). Third, positions of the L414 and N415 side chains in ΔASIC1/JNJ–799760 are "un-swapped" (as seen in closed-state structures; Supplementary Table 1 and Supplementary Fig. 5), as opposed to the swapped positions in

**Table 1 Crystallographic data collection and refinement statistics.**

|  | ΔASIC1/JNJ-799760[a] |
|---|---|
| *Data collection* |  |
| Space group | P2₁2₁2₁ |
| Cell dimensions |  |
| *a, b, c* (Å) | 91.79, 116.15, 227.79 |
| *α, β, γ* (°) | 90, 90, 90 |
| Resolution (Å) | 3.0 (3.1–3.0)[b] |
| $R_{sym}$ or $R_{merge}$ | 0.153 (0.783) |
| $I/\sigma I$ | 11.1 (1.7) |
| Completeness (%) | 98.10 (87.39) |
| Redundancy | 5.0 (3.6) |
| *Refinement* |  |
| Resolution (Å) | 35.2–3.0 |
| No. of reflections | 48,177 (4219) |
| $R_{work}/R_{free}$ | 0.224/0.267 |
| No. of atoms |  |
| Protein | 9697 |
| Ligand/ion | 130 |
| Water | 0 |
| *B*-factors |  |
| Protein | 79.98 |
| Ligand/ion | 90.71 |
| Water | - |
| R.m.s. deviations |  |
| Bond lengths (Å) | 0.011 |
| Bond angles (°) | 1.35 |

[a]One crystal was used for data collection.
[b]Values in parentheses are for the highest-resolution shell.

3S3W (Fig. 9d) and other desensitized structures (Supplementary Table 1). The un-swapping involves a 6.9 Å swing of L414 away from the central vestibule and a significant rearrangement of the β11–β12 linker in the palm domain, consistent with the critical roles that these residues play in channel desensitization and recovery[30,33]. Fourth, the second transmembrane domain (TM2) helix of ΔASIC1/JNJ–799760 undergoes a (TM2b) domain swap that allows the Gly443–Ala444–Ser445 motif (a.k.a. the GAS belt) to adopt an extended conformation, as that adopted by closed-state structures (Supplementary Table 1 and Supplementary Fig. 5). In contrast, the TM2 domain of 3S3W is continuous with no GAS belt extension or TM2b domain swap (Fig. 9e). Finally, the channel gate of ΔASIC1/JNJ–799760 is shut (as in closed-state structures), in contrast to the (paradoxical) "open" gate in 3S3W (Supplementary Table 1 and Supplementary Fig. 5).

The closed gate in ΔASIC1/JNJ-799760 rules out the possibility of an open or desensitized-like channel. Remarkably, the conformational features of ΔASIC1/JNJ-799760 described above are adopted invariably by all closed-state structures (Supplementary Table 1). Only two (closed gate and TM2b domain swap) of the five conformational features are shared between closed and desensitized structures, whereas the other three (expanded acidic pocket, flipped T84–R85 bond and non-swapped L414–N415 side chains) are characteristic only of closed channels, demonstrating that ΔASIC1/JNJ-799760 is a closed-state structure.

Additionally, ion binding data further support this contention. In structures of cASIC1 channels in the open[20] and desensitized[3,21,25], but not closed[18,32], states, a Cl⁻ ion is bound to a site coordinated by R310 and E314 from one subunit and by K212 from a neighboring subunit. There is no evidence of Cl⁻ binding to this site in the structure of ΔASIC1/JNJ-799760.

Taken together, JNJ-799760 stabilizes the closed state of the channel, as demonstrated by both functional and structural results.

**JNJ-67869386 and NJ799760 occupy the same binding pocket.** Though structurally distinct, JNJ-67869386 and JNJ-799760 interact functionally in a way consistent with their binding competitively to the same site. As shown in Fig. 10a, the time course of current recovery from co-inhibition by JNJ-799760 and JNJ-67869386 lies intermediate between the kinetics of recovery individually from each of the molecules and is biphasic with characteristic time constants for the two molecules. This profile is exactly what would be expected if binding of the two molecules is mutually exclusive. If JNJ-799760 and JNJ-67869386 bound independently to distinct sites, current recovery would be dominated by the kinetics of the molecule that dissociates from the channel more slowly, JNJ-67869386, contrary to what is observed here.

To understand the molecular interactions of JNJ-67869386 with the channel, we developed a binding model based on the crystal structure of ΔASIC1/JNJ-799760. We hypothesized that both compounds occupy the same site (based on the data in Fig. 9a and their similar functional profiles). Molecular docking of JNJ-67869386 to the JNJ-799760-binding site in ΔASIC1/JNJ-799760 provides a model consistent with the molecules binding at the same site (Fig. 10b). The main hydrogen bond is formed between JNJ-67869386 and the backbone nitrogen of V327, an interaction also made by JNJ-799760. The amino moiety of the amino-indazole in JNJ-67869386 makes an additional hydrogen bond to the backbone carbonyl of C336. An equivalent interaction is not present in ΔASIC1/JNJ-799760. The core of JNJ-67869386 makes aromatic and hydrophobic interactions with the side chains of F99, L115, L116, Y159, M326, V327, and Y341, similar to those that JNJ-799760 makes with these residues in the crystal structure. Due to its shorter length, JNJ-67869386 does not form direct hydrogen bonds with L116 or P232 as JNJ-799760 does, although there may be a water-bridged interaction with P232. Overall, JNJ-67869386 makes interactions with many of the same residues that JNJ-799760 interacts with. Importantly, all the residues that JNJ-678969386 and JNJ-799760 interact with in the model or the crystal structure are conserved between cASIC1 and rASIC1a.

The docking model and co-crystal structure produce predictions regarding the interactions of JNJ-67869386 and JNJ-799760 with the surrounding channel residues that are testable by site-directed mutagenesis. Based on these interactions, we studied the effect of these molecules on two rASIC1a mutants, F98A and Y340A (corresponding to F99A and Y341A in cASIC1, respectively). The side chains of both F99 and Y341 of cASIC1 make aromatic interactions with JNJ-799760 in the crystal structure and with JNJ-678969386 in our binding model. Removing the aromatic side chain by mutating to alanine is predicted to decrease the potency of the compounds. Indeed, the potency of inhibition of pH 6.0-induced current by JNJ-799760 and JNJ-67869386 is significantly decreased for both mutants as predicted (Fig. 10c), providing further functional validation of the co-crystal structure and docking model. The pH dependence of activation and steady-state desensitization is similar for the mutant and wild-type channels (Supplementary Fig. 6), making the pharmacological comparisons straightforward as the same holding and test pHs can be (and are) used for all three channels. Taken together, these results indicate that the two molecules interact similarly and predictably with the two amino acid residues, strongly supporting the contention that they bind to the same pocket.

## Discussion

Small molecules have been shown to variously modulate ASIC1a. For example, spermine decreases desensitization and current

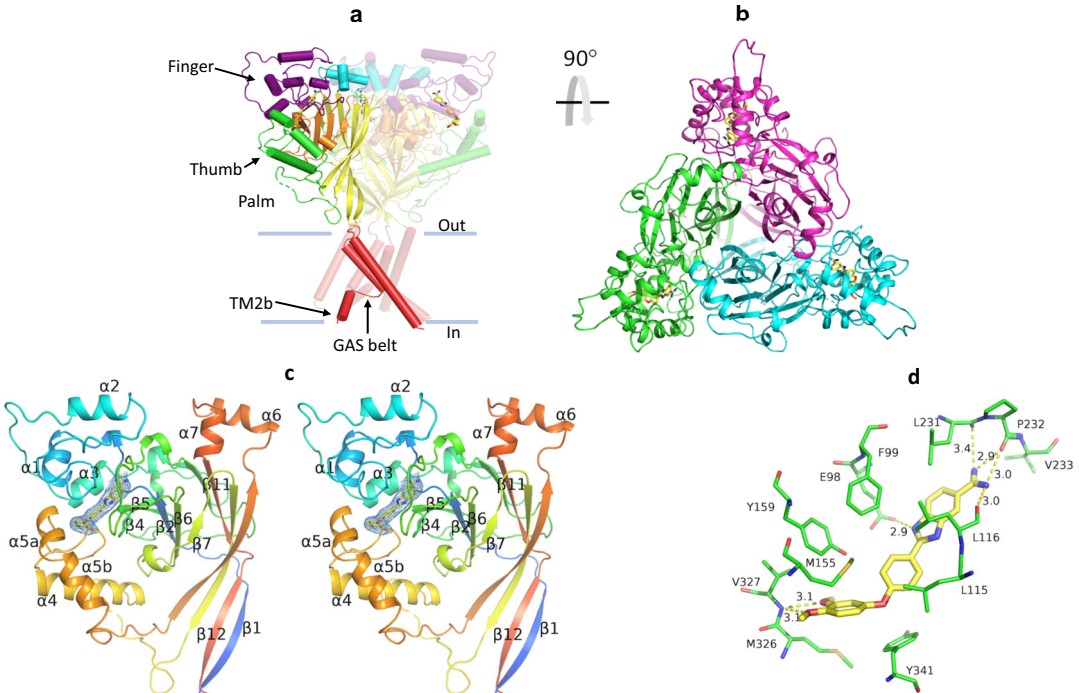

**Fig. 8 Crystal structure of the ΔASIC1/JNJ-799760 complex. a** Architecture of ΔASIC1/JNJ-799760 viewed parallel to the membrane. One subunit is highlighted with a different color for each domain. JNJ-799760 is in yellow. **b** The same structure viewed extracellularly with each subunit in a different color. **c** Stereo view of the JNJ-799760-bound ECD (chain A). Key α helixes and β sheets are labeled. The 2Fo−Fc electron density map of JNJ-799760 contoured at 1.0σ is shown as blue mesh. **d** Stick representation of JNJ-799760 and the surrounding channel residues in the binding pocket (chain A). Numbers next to the dotted lines indicate distances (in Å). Oxygen and nitrogen atoms are shown in red and blue, respectively.

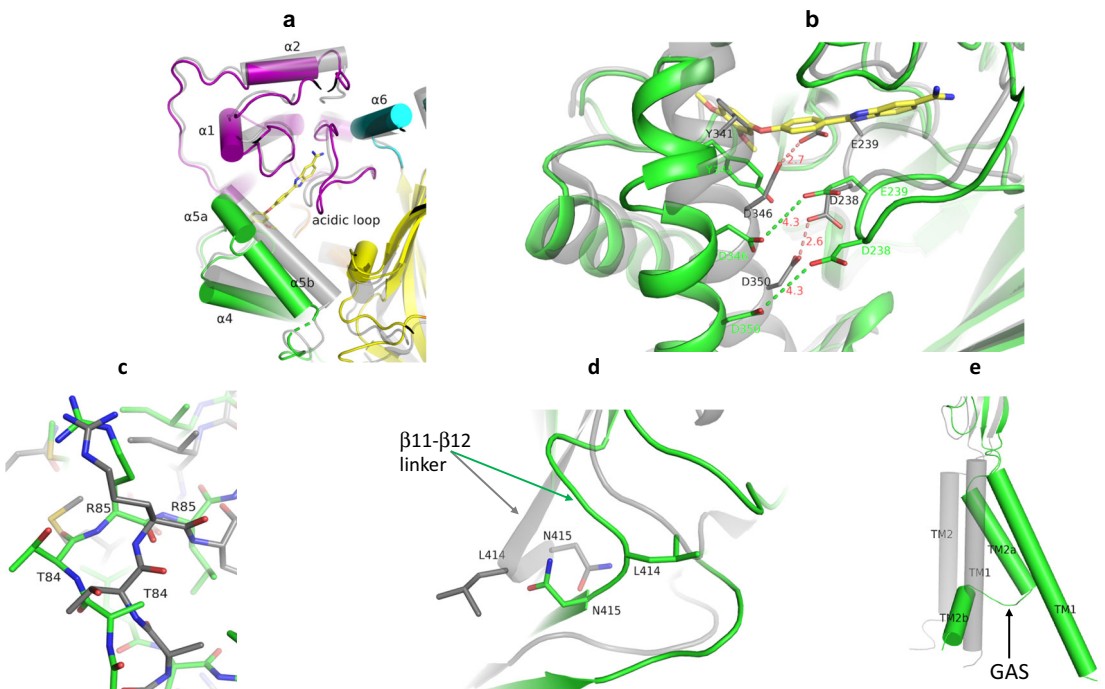

**Fig. 9 Key conformations of the ΔASIC1/JNJ-799760 complex. a** Superposition of ΔASIC1/JNJ-799760 (colored by domain) and 3S3W apo (gray/transparent) structures. Binding of JNJ-799760 causes outward pivots of α4 and α5 (split into α5a and α5b) and displacement of the acidic loop. **b** Superposition of ΔASIC1/JNJ-799760 (green) and 3S3W (gray/transparent) near the acidic pocket. Note the increased side-chain distance of D238–D350 and of E239–D346. Numbers are in Å. Note that E239 and Y341 side chains in 3S3W clash with JNJ-799760. **c** Superposition of ΔASIC1/JNJ-799760 (green) and 3S3W (gray) for a portion of the β1–β2 linker. Note the ~180° flip of the T84–R85 peptide bond orientation. **d** Superposition of ΔASIC1/JNJ-799760 (green) and 3S3W (gray) for a portion of the β11–β12 linker. Note the swap of L414–N415 side-chain orientations and displacements of L414 and β11–β12 linker due to JNJ-799760 binding. **e** Comparison of chain A TM domains of ΔASIC1/JNJ-799760 (green) and 3S3W (gray). Note the extended GAS belt conformation and TM2b domain swap in ΔASIC1/JNJ-799760.

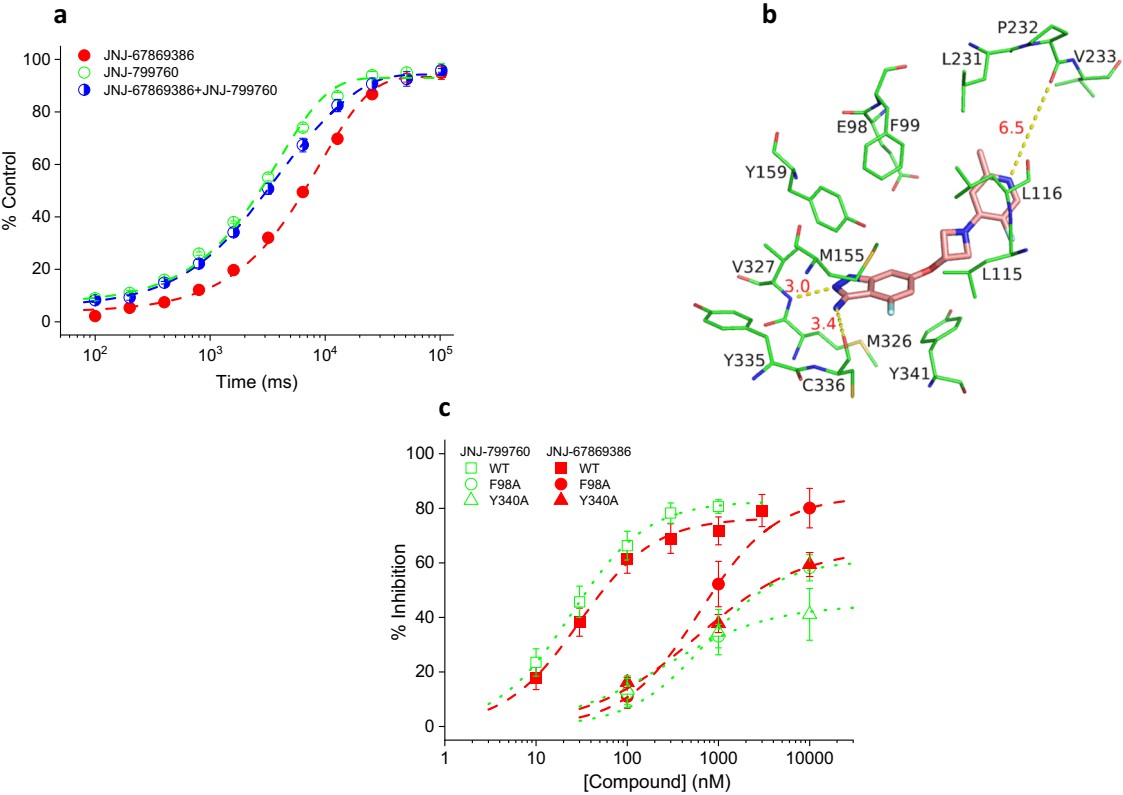

**Fig. 10 JNJ-67869386 and JNJ-799760 bind to the same pocket. a** Kinetics of current recovery from inhibition by 100 nM JNJ-67869386, 1 μM JNJ-799760 or 100 nM JNJ-67869386 + 1 μM JNJ-799760. The steady-state current in the presence of compound is subtracted before normalization. Data are fitted to a single or double exponential function (dashed lines) with time constants of 9.1 ± 0.4 s ($n = 11$; solid circles), 3.1 ± 0.2 s/14.9 ± 1.6 s ($n = 5$; half-filled circles) and 3.9 ± 0.2 s ($n = 5$; open circles), respectively, for JNJ-67869386, JNJ-67869386 + JNJ-799760, and JNJ-799760. The three groups of data are significantly different from one another ($p < 0.001$; Two-way ANOVA). The holding pH is 8.2. **b** Docking of JNJ-67869386 (salmon carbons) to the binding site of JNJ-799760 on ΔASIC1 (green carbons). Stick representation of JNJ-67869386 and channel residues with which it interacts. Specific interactions between JNJ-67869386 and ΔASIC1 are shown as yellow dotted lines labeled with distances in Å. **c** Concentration-dependent inhibition of pH 6.0-induced currents of rASIC1a WT, F98A, and Y340A channels by JNJ-67869386 (solid symbols) and JNJ-799760 (open symbols). Data are fitted to a logistic function (dashed and dotted lines). For JNJ-67869386, $IC_{50} = 28.9 \pm 3.6$ nM ($n = 7$), 626.3 ± 11.0 nM ($n = 4$) and 571.8 ± 176.9 nM ($n = 5$), respectively. For JNJ-799760, $IC_{50} = 24.9 \pm 1.3$ nM ($n = 4$), 725.9 ± 356.1 nM ($n = 4$) and 260.0 ± 89.8 nM ($n = 4$), respectively. All mutant channel data are significantly different from the corresponding WT ($p < 0.001$; two-way ANOVA).

inhibition by PcTx1[16,34]; compound 5b, a PcTx1-inspired small synthetic molecule that is modeled to bind in the acidic pocket, is an allosteric inhibitor of channel activation[28]; Daurisoline and 2-guanidine-4-methyl-quinazoline (GMQ, a non-proton agonist of ASIC3) both cause an acidic shift of the pH dependence of ASIC1a activation and desensitization[35,36]; and histamine shifts the pH dependence of activation toward more basic pHs and of desensitization toward more acidic pHs[37,38]. However, these and other studies of small molecule modulation of ASIC1a to date have generally lacked substantive evidence necessary for elucidating the molecular and structural basis of modulation. As of this study, amiloride was the only small molecule co-crystalized with ASIC1. Aside from binding to and plugging the channel pore, it also binds to the acidic pocket, to a site distinct from JNJ-799760 but overlapping with PcTx1[20]. However, the functional relevance of this binding is unknown.

In this study, we combine functional, structural, computational, and mutational approaches to elucidate a mechanism by which two chemically distinct small molecules modulate the gating of ASIC1a. We show that structurally, both molecules bind to the same and previously unrecognized allosteric site at the acidic pocket. The crystal structure of ΔASIC1/JNJ-799760 reveals that JNJ-799760 keeps the acidic pocket in an expanded conformation as well as the channel in an overall conformation

commensurate with the closed state. Functionally, these molecules serve as both negative and positive modulators. As negative modulators, they inhibit H⁺-evoked currents by shifting the pH dependence of activation toward more acidic pHs. As positive modulators, they impede channel desensitization and tachyphylaxis by causing an acidic shift of the pH dependence of steady-state desensitization. As such, they are gating modifiers that stabilize the closed state, corroborating the conclusion from the structural studies. Our results identify a previously unknown drug-binding site and represent a direct structural demonstration that binding of a small molecule modulates the gating of an ASIC channel.

JNJ-67869386 and JNJ-799760 bind to a previously unrecognized site at the acidic pocket, in a conformation distinct from what PcTx1 or amiloride interacts with[19,20,25]. Binding of different chemotypes of small molecules to this site highlights the significance of the locus for pharmacological modulation.

The acidic pocket contains three pairs of conserved acidic amino-acid residues. In the closed state of the channel, these pairs are deprotonated. The resulting electrostatic repulsion is thought to be responsible for maintaining the acidic pocket in an expanded conformation[18]. Protonation of these residues in the open and desensitized states removes the electrostatic repulsion, causing the acidic pocket to collapse. JNJ-799760 helps to keep

these pairs apart via extensive interactions with nearby residues. The resulting expanded acidic pocket requires higher $H^+$ concentrations to collapse, leading to an acidic shift in the pH dependence of channel activation.

Functional studies indicate that JNJ-799760 and JNJ-67869386 do not bind to desensitized channels at equilibrium. First, JNJ-799760 and JNJ-67869386 cannot bind to pre-desensitized channels, consistent with the notion that the compound binding site is either inaccessible, distorted or non-existent with the acidic pocket being in a collapsed conformation. Second, closed channels do not desensitize with compound bound. Instead, compound must first dissociate from the channel before desensitization can occur. Third, although open channels can initially undergo desensitization with compound bound, the compound-bound desensitized states (OBD in Fig. 6a) are unstable and channels irreversibly transition to compound-unbound desensitized states (OD).

Our X-ray data provide a structural basis for this mechanism. Structural overlays show that the side chain positions of E239 and Y341 in both 3S3W (Fig. 9b) and 2QTS (Supplementary Fig. 7) are in direct steric clash with JNJ-799760 in our co-crystal structure. This mutual exclusivity/occlusion indicates that the conformation of the binding pocket in the desensitized state is incompatible with JNJ-799760 binding. A similar comparison with 3S3X, a structure of the ΔASIC1/PcTx1 complex at pH 5.5[25], shows the existence of the same incompatibility (Supplementary Fig. 8). In addition, binding of JNJ-799760 pushes the α4 and α5 helices, with which PcTx1 makes extensive contacts, away from the acidic loop compared to the PcTx1-bound conformation (Supplementary Fig. 8). These structural conflicts are also evident in an overlay of 3S3X with the JNJ-67869386-docked pose (Supplementary Fig. 8), in agreement with the kinetic and steady-state data on the functional antagonism of the PcTx1 effect by JNJ-67869386.

Beyond the acidic pocket, binding of JNJ-799760 also produces large, global changes in the channel conformation, including (1) causing an about-face flip of the T84–R85 peptide bond in the β1–β2 linker of the palm domain, (2) un-swapping the side-chain positions of L414–N415 in the β11–β12 linker of the palm domain, (3) inducing the TM2b domain swap and extended conformation of the GAS belt, and (4) shutting the channel gate. These are also conformations invariably adopted by closed-state structures (Supplementary Table 1), suggesting that binding of JNJ-799760 destabilizes the desensitized-like state of the apo structure in favor of the closed state, which corroborates the conclusions based on our functional studies. These results, along with the fact that the predictions of our structural analysis are borne out by the functional studies of mutant channels, provide strong evidence of structure–function correlation.

As with JNJ-799760 and JNJ-67869386, divalent cations, such as $Ca^{2+}$, $Ba^{2+}$, and $Mg^{2+}$, also stabilize the closed state of ASIC1a[34]. Yoder et al.[32] showed that binding of these ions at certain extracellular sites in ASIC1 is state dependent—occupancy is observed in the closed, but not in the desensitized state. These findings suggest a degree of similarity between modulation by divalent cations and our molecules. It is worth noting, however, that binding of divalent cations to these sites does not corelate with functional stabilization of the closed state, in contrast to what we observe for JNJ-799760 and JNJ-67869386.

Although a crystal structure of the ASIC1/JNJ-67869386 complex is lacking, several lines of evidence strongly indicate that JNJ-67869386 binds to the same pocket as JNJ-799760. First, current recovery from simultaneous inhibition by the two compounds takes on an intermediate, biphasic time course with characteristic time constants for the two individual molecules. This indicates that binding of these molecules is mutually exclusive rather than independent, the latter of which would, contrary to observation, result in the time course being dominated by the slower recovery from JNJ-67869386. Second, docking of JNJ-67869386 to the JNJ-799760 site produces many of the same interactions with the surrounding channel residues as JNJ-799760 does in ΔASIC1/JNJ-799760. Third, mutations of residues expected to interact with both JNJ-799760 (based on the crystal structure) and JNJ-67869386 (based on the docking model) significantly and similarly change the potency of JNJ-799760 and JNJ-67869386 and in the same direction as predicted. Lastly, effects of JNJ-67869386 and JNJ-799760 on ASIC1a function are qualitatively identical.

Given that the acidic pocket also adopts a collapsed conformation in the open state[20], it is possible that at equilibrium, our molecules do not bind to the open state either. Consistent with this scenario, open-channel desensitization is faster in the presence of these compounds, suggesting compound-induced destabilization of channel opening. However, we cannot address this question directly in the current study because (1) structural information on an open channel in complex with our compounds is lacking (Indeed, whether our compounds can be co-crystalized with an open channel would itself constitute a direct test of this hypothesis), and (2) the rate of dissociation of these molecules from wild-type ASIC1a is too slow relative to the duration of channel openings to observe their dissociation directly from the open channel. Experiments using a mutant channel with long open durations should help test this hypothesis functionally.

Of the amino acid residues in ASIC1a that interact with JNJ-799760, only three are different in ASIC2a. Amino acids E97, Y158, and Y340 in rASIC1a are Gly, Leu, and His, respectively, at the corresponding positions in rat ASIC2a. However, JNJ-799760 and JNJ-67869386 are inactive at ASIC2a ($IC_{50} > 50\,\mu M$), suggesting that one or more of these residues may be critical for the ASIC1a selectivity. Alternatively, the apparent selectivity of these compounds may be a consequence of diminished affinity and/or efficacy at high proton concentrations necessary to activate ASIC2a. Additional experiments with ASIC2a mutated to the corresponding ASIC1a amino acids at these positions may help to distinguish between these hypotheses.

In this study, we identified a small molecule binding site in ASIC1 and presented direct structural evidence that small molecule binding modulates the gating of an ASIC channel. Furthermore, we elucidated the molecular mechanism and structural basis of this modulation. Our findings provide important mechanistic and structural insight into the modulation of ASIC channels and contribute to the understanding of structure, function, and therapeutic targeting of this class of ion channels.

## Methods

**Cell culture and transient transfections**. Chinese Hamster Ovary (CHO) cells stably expressing rat ASIC1a were cultured in Ham's F12, supplemented with 10% FBS, 1% penicillin–streptomycin and 500 μg/mL G418 and incubated at 37 °C with 5% $CO_2$. Transient transfections of rASIC1a and cASIC1 wild-type or mutant channels in CHO cells were performed with Lipofectamine 3000 (Thermo Fisher Scientific) according to the manufacturer's recommendations. The green fluorescent protein (GFP) cDNA (at 1/10 the amount of channel cDNA) was co-transfected with channel cDNA to aid the identification of transfected cells under the microscope in patch clamp experiments. All cDNAs were sequence verified.

**Electrophysiology**. Cells were freshly dissociated with CellStripper (Corning) and dispersed in a chamber on the stage of an inverted microscope. Upon formation of the whole-cell conformation, the cell was lifted from the bottom of the chamber and placed at the tip of a tubing perfusing an extracellular solution containing (in mM): 149 NaCl, 2 $CaCl_2$, 4 KCl, 1 $MgCl_2$, 5 glucose, 10 HEPES, pH 7.4, 310 mOsm/L. Extracellular solutions at more basic or acidic pHs were made by titrating the above pH 7.4 solution with NaOH or HCl (5 mM MES was added to solutions at pH 6.0 and lower). Pipette electrodes were filled with an intracellular solution

containing (in mM): 135 KCl, 4 MgATP, 0.3 Na$_2$GTP, 10 EGTA, and 20 HEPES, pH 7.2, 290 mOsm/L.

All recordings were performed at room temperature using an Axopatch 200B amplifier and pClamp 11 software (Molecular Devices). Currents were measured by whole-cell patch clamp, digitized at 10 kHz and lowpass filtered at 2 kHz. Series resistance was 75% compensated. Responses were elicited by rapid perfusion of acidic solutions using the SF-77B Fast-Step Perfusion device (Warner Instruments) for 40 ms once every 30 s (unless indicated otherwise) and recorded till steady state was reached. The holding potential was 0 mV unless indicated otherwise.

PcTx1 was purchased from Peptides International (Louisville, Kentucky, USA) and solubilized in aqueous buffer as 1 mM stock. JNJ-67869386 and JNJ-799760 were synthesized in house and solubilized in DMSO as 10 mM stocks. All other chemicals were from Tocris (Minneapolis, USA).

**Protein expression and purification**. To form well-ordered crystals, ΔASIC1 with a N-terminal FLAG tag, a 10 × His tag and TEV protease cleavage site was expressed in Sf9 cells using the Bac-to-Bac baculovirus expression system. To prepare crude cell membranes, the cell pellet was first resuspended in a lysis buffer (20 mM KCl, 10 mM MgCl$_2$, 10 mM HEPES, pH 7.5) containing the cOmplete™ EDTA-free protease inhibitor cocktail (1 tablet/50 mL; SigmaAldrich). After passing through a microfluidizer three times at 600 kPa, the lysate was centrifuged at 45,000 rpm (45Ti rotor) for 30 min. The pellet was then homogenized in a high salt buffer (lysis buffer containing 1 M NaCl) and centrifuged again at 45,000 rpm for 30 min. Finally, the crude membrane pellet was re-homogenized in a freezing buffer (lysis buffer containing 40% glycerol) and stored at −80 °C. For protein purification, crude membranes were first solubilized on ice for 2 h in a buffer containing 20 mM Tris (pH 7.5), 20 mM imidazole (pH 7.5), 150 mM NaCl, 10 mM MgCl$_2$, 5 mM beta-mercaptoethanol, 2% DDM, 10% glycerol, and the cOmplete™ EDTA-free protease inhibitor cocktail. Unsolubilized elements were removed by centrifugation at 45,000 rpm for 30 min. The supernatant was incubated at 4 °C overnight with washed Talon resin. The resin was then thoroughly washed first with washing buffer (20 mM pH 7.5 Tris, 35 mM pH 7.5 imidazole, 150 mM NaCl, 10 mM MgCl$_2$, 5 mM beta-mercaptoethanol, 0.05% DDM, 10% glycerol), and then again with washing buffer containing 100 µM JNJ-799760 (diluted from a 50 mM stock in 100% DMSO). The compound-bound ΔASIC1 protein was eluted with elution buffer (washing buffer containing 250 mM imidazole and 100 µM JNJ-799760). To remove the FLAG and His tags, TEV protease was added to the eluted protein (TEV protease:ΔASIC1 = 1:10, w–w). After incubation at 4 °C overnight, the sample was centrifuged at 20,000×g for 15 min to remove any precipitation and then applied through a Superdex 200 column in a solution containing 10 mM Tris (pH 7.5), 150 mM NaCl, 0.05% DDM, 5 mM beta-mercaptoethanol, 1 mM EDTA, and 25 µM JNJ-799760. The major peak fraction containing highly purified ΔASIC1 was collected, concentrated to 4 mg/mL using a Vivaspin® 6 concentrator (MWCO 100 kDa; Sartorius) at 2000×g, frozen in liquid nitrogen and stored at −80 °C.

**Crystallization**. Before crystallization, spermine and JNJ-799760 were added to protein samples to a final concentration of 10 mM and 100 µM, respectively. Crystals were obtained at 13 °C using the hanging drop vapor diffusion method. Drops were set up by mixing 1 µL protein sample and 1 µL reservoir solution (100 mM MgCl$_2$, 100 mM HEPES, 28–30% PEG 400). Crystals were frozen and stored in liquid nitrogen for subsequent data collection.

**X-ray data collection, processing, and structure determination**. X-ray diffraction data were collected at 100 K and 1.0 Å wavelength on LS-CAT beamline 21-ID-D with a MAR 300 CCD detector (Argonne National Laboratory, Illinois, USA). Data frames were indexed and diffraction spots were integrated and scaled using the HKL-2000 program package[39]. The structure of the ΔASIC1/JNJ-799760 complex was determined by molecular replacement using 3S3W as a search model. Models were built with iterative rounds of manual model building in Coot[40] and refined in REFMAC[41] from the CCP4 program suite[42] and Phenix.refine in the Phenix software[43] until satisfactory statistics were achieved. The diffraction data processing and structure refinement statistics are summarized in Table 1. The crystal structure displays Ramachandran statistics with 95.78% of residues in the most favored regions and 4.06% of residues in the allowed regions of the Ramachandran diagram. The final refinement statistics and geometry of the crystal structure are shown in Table 1.

**Modeling of JNJ-67869386 binding to ΔASIC1**. We developed a binding model for JNJ-67869386 based on the crystal structure of ΔASIC1/JNJ-799760 and the hypothesis that both molecules occupy the same site. JNJ-67869386 was subject to automated docking using Schrödinger's Glide module (v2017-1)[44–47]. The binding site grid was generated with default parameters, centered on the ligand in chain B. JNJ-67869386 was prepared with LigPrep and docked using Glide SP, all with default parameters.

**Kinetic modeling**. Kinetic simulations were performed using ChanneLab (Synaptosoft, Decatur, GA), with the following assumptions: (1) subunits are independent/non-cooperative, (2) channels with at least one desensitized subunit

do not conduct current[33]; (3) partially and fully compound-bound open states are activated with the same probability and conductance; and (4) kinetics of compound binding/unbinding are the same at pH 8.2 and pH 7.1. Kinetics were simulated with the rate constants δ, θ, α, and β adjusted to produce the best visual global (simultaneous) fits to the experimental data at holding pHs of 8.2 (compound binding/unbinding and recovery from desensitization) and 7.1 (onset of closed-state desensitization). Since the channel opening/closing kinetics are much faster than that of compound binding/unbinding, the fractional current amplitude (simulating the experimental current response elicited by a pH 6.0 test pulse) was calculated as follows: $I(t) = \gamma_O \times Q_O \times P_C(t) + \gamma_{OB1} \times Q_{OB1} \times P_{CB1}(t) + \gamma_{OB2} \times Q_{OB2} \times P_{CB2}(t) + \gamma_{OB3} \times Q_{OB3} \times P_{CB3}(t)$, where $t$ denotes time; the subscripts denote open (O, OB$_1$, OB$_2$, and OB$_3$) or closed (C, CB$_1$, CB$_2$, and CB$_3$) states with zero, one, two, or three molecules bound; $\gamma$ denotes conductance relative to the compound-unbound open state (thus, $\gamma_O \equiv 1$); $P(t)$, the simulated response, represents the channel occupancy in each closed state as denoted by the subscript; and $Q$ represents the conditional probability of channel opening (by a pH 6.0 test pulse) given that the channel is in the corresponding closed state (from which it is activated). Partly based on estimations from pH responses, the following values were used in the simulations: $\gamma_{OB1} = \gamma_{OB2} = \gamma_{OB3} = 0.77$, $Q_O = 0.7$, $Q_{OB1} = Q_{OB2} = Q_{OB3} = 0.15$.

**Electrophysiology data analysis**. Baseline values (i.e., current amplitudes at the conditioning pH) were subtracted to obtain responses evoked by the test pH. Responses were normalized for each cell before averaging (see figure legends for more detail on normalization for each type of experiment). Concentration–response data were fitted to a logistic function of the form: $R = (A_1 - A_2)/(1 + (C/C_0)^h) + A_2$, where $R$ is the normalized response, $C$ is either pH or compound concentration, $C_0$ is the pH/concentration at which half-maximal response occurs (pH$_{50}$ or IC$_{50}$), $h$ is the Hill coefficient, and $A_1$ and $A_2$ are constants. Kinetic parameters were obtained by fitting the data with either a single or double exponential function. Fitted data are shown as solid, dashed, or dotted curves.

**Statistics and reproducibility**. Statistical analyses were performed using two-tailed Student's $t$-test, or one- or two-way ANOVA with post-hoc Tukey test as described in the text. Experimental results are reproducible and reported as mean ± SEM over independent measurements on n different cells. Data fitting and statistical analyses were performed using Origin (Northampton, MA, USA).

**Reporting summary**. Further information on research design is available in the Nature Research Reporting Summary linked to this article.

## Data availability
Coordinates of ΔASIC1/JNJ-799760 were deposited to the online database https://www.rcsb.org/ with accession code 6X9H[48]. The source data underlying the graphs and charts presented in the main figures are provided in Supplementary Data 1. The other datasets generated and/or analyzed during the current study are available from the corresponding author upon reasonable request.

## Code availability
All software used for this study are commercially available.

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

## Author contributions

Y.L. conceived, designed, and carried out the electrophysiological studies and data analysis as well as performed the kinetic modeling. J.M. and B.G. designed and J.M. carried out the crystallography studies. J.M., R.L.D., and J.L. performed the data analysis. J.R. and Y.L. helped with the initiation of the crystallography studies. R.L.D. planned and carried out the molecular modeling studies, which contributed to compound design. D.L., J.R., and M.L. were involved in designing/synthesizing JNJ-799760 and JNJ-67869386. J. S. conducted the initial functional studies confirming the compounds' effects. R.H. maintained cell cultures and contributed to the early functional confirmation of the compounds' effects. C.L. generated the channel plasmids. R.M. was involved in the expression of ΔASIC1 for crystallography. Y.L., J.M., R.L.D., and M.M. wrote the manuscript. All authors contributed to reviewing and revising the manuscript.

## Competing interests

At the time of the work described in this manuscript, all authors were employees of Janssen Research & Development, LLC, a division of Johnson & Johnson, the funder of this study.
