## [Peer Review File · Communications Biology]

Reviewers' comments:

Reviewer #1 (Remarks to the Author):

The paper contains interesting information, but has shortcomings in the description of the history of ASICs discovery and exploration.

ASICs were discovered in 1980 (as "the receptor of rprotons"). The fact is quite well known, but this does not mean that it should not be acknowledged.

The use of small molecules binding to the acidic pocket of ASIC molecule is not pioneered by the authors. At least one successful attempt has been already described (Buta et al., J Med Chem, 2015) and should also be referred.

Extensive studies with small molecules vs ASICs performed by Abbott and Merck groups has been also neglected.

After reading this Md general picture of the field looks as follows: "amiloride, psalmotoxin and present study".

Reviewer #2 (Remarks to the Author):

In this work, the authors show that two potent allosteric effectors of an acid-sensing ion channel work by stabilising the closed state.

They present extensive electrophysiology experiments as well as a crystal structure of a complex of one of them JNJ-799760 with the channel.

As my background is more on structural biology I will focus my comments on this aspect of the paper. Essentially, I think the paper is mostly fine, but I would suggest to change the presentation of the structural results, which are confined to only one Figure (Figure 7), out of 8. This Figure contains 9 panels a-i and it is to be expected that all of them will be unreadable in the final version of manuscript. I therefore recommend to split it into 2 or 3 Figures, highlighting the panels with the electron density.

I also have a remark on Fig. 5e, the Kinetic Model: this is interesting but what does the associated Energy landscape look like?

See also p. 20 "It is energetically un favorable for compound to remain bound in D form....": could the authors justify this statement?

In the Materials and Methods Section, the authors should state how they dissolved the compounds, because these compounds look very hydrophobic so it was probably pure DMSO or some other organic solvent. In that case, we need to know the concentration of the stock solution, and the final concentration of the organic solvent in the crystallisation drops. Probably useful also to know if one wants to reproduce the electrophysiology experiments.

Minor points

Typos: the authors keep putting no space between pH and its value, as in pH6.0 instead of pH 6.0

-p. 14 "Relatvie" -> Relative

-p. 25 What is the "viper diffusion method" (sic)? Probably vapour diffusion method?

-p. 26 Scordinger -> Schrodinger

Reviewer #3 (Remarks to the Author):

This paper by Liu et al describes experimental and structural data for a small molecule modulator of the ASICs channel. The topic is important, the study is carefully carried out and the data are convincing.

In my view the study contains four distinct parts:

1. Kinetic characterization of the effects of two compounds on the channel.
2. A kinetic model.
3. X-ray structure for the channel with bound molecules.
4. Characterization of the binding site.

While I think that parts 1, 3 and 4 are carefully considered, point 2 only provides a qualitative and fairly vague, not very informative model. A quantitative but still simple model would greatly improve the understanding of the experimental data and would (probably) be a convincing argument in the characterization of the binding site. I strongly suggest the authors to implement a quantitative kinetic model.

Do your binding pocket explain the drug effect on the pH dependence? I guess so, but I missed the explanation. Please, clarify.

Minor points:

Page 5: remove the sentence about preliminary data. Data not shown is not OK.

Page 12: the relation between tachyphylaxis and desensitization is not clear to me. Try to explain this better.

Responses to referee #1:

Reviewer:

ASICs were discovered in 1980 (as "the receptor of protons"). The fact is quite well known, but this does not mean that it should not be acknowledged.

Reply:

Indeed. We have now included this reference (Krishtal and Pidoplichko, 1980) in the revised manuscript (the new ref. #1).

Reviewer:

The use of small molecules binding to the acidic pocket of ASIC molecule is not pioneered by the authors. At least one successful attempt has been already described (Buta et al., J Med Chem, 2015) and should also be referred.

Reply:

We cited Buta et al. (2015) in several places in the original manuscript except in Introduction. We have now referenced this study in Introduction as well in the revised version.

Reviewer:

Extensive studies with small molecules vs ASICs performed by Abbott and Merck groups has been also neglected.

Reply:

Dubé et al. (2005), who reported a novel small molecule ASIC channel blocker with efficacy in pain models, was cited in the Introduction of the original version (ref #8), albeit only in the context of biological processes in which ASIC1a is implicated. We have now explicitly indicated in a later paragraph (also in Introduction) that this is a novel, non-amiloride-like small molecule blocker of ASIC channels. It should be noted that no mechanistic information about this molecule is reported.

Responses to referee #2:

Reviewer:

I would suggest to change the presentation of the structural results, which are confined to only one Figure (Figure 7), out of 8. This Figure contains 9 panels a-i and it is to be expected that all of them will be unreadable in the final version of manuscript. I therefore recommend to split it into 2 or 3 Figures, highlighting the panels with the electron density.

Reply:

Good suggestion. The structural data are now spread in two figures, the new Fig. 8 (a-d) and Fig. 9 (a-e).

Reviewer:

I also have a remark on Fig. 5e, the Kinetic Model: this is interesting but what does the associated Energy landscape look like? See also p. 20 "It is energetically unfavorable for compound to remain bound in D form....": could the authors justify this statement?

Reply:

The old Fig. 5e (qualitative kinetic model) is now part of the new Fig. 6 (Fig. 6a), in which we have implemented a simple quantitative model that explicitly does away with compound-bound desensitized states (e.g., CBD in the new Fig. 6a). This simple model produced good global fits to the kinetic data, lending further and quantitative support to our contention that the compounds do not bind to desensitized states at equilibrium. We have also added a corresponding paragraph each in the Results and Methods sections describing the modeling work.

Based on the kinetic modeling results, we made a slight modification to the qualitative kinetic model. Specifically, the fact that closed-state desensitization in the presence of compound can be quantitatively accounted for by the $CB \rightarrow C \rightarrow CD$ transitions (new Fig. 6c, d and f) indicates that the $CB \rightarrow CBD$ transition does not occur. Alternatively, it could be said that the rate of $CBD \rightarrow CB$ transition is \gg that of $CB \rightarrow CBD$, such that the probability of being in the CBD state is essentially zero relative to that of being in the CB state (i.e., an effectively irreversible transition).

We removed the statement: "It is energetically unfavorable for compound to remain bound to desensitized channels" and only kept the second half of the sentence stating the irreversible nature of compound dissociation in desensitized states.

Reviewer:

In the Materials and Methods Section, the authors should state how they dissolved the compounds, because these compounds look very hydrophobic so it was probably pure DMSO or some other organic solvent. In that case, we need to know the concentration of the stock solution, and the final concentration of the organic solvent in the crystallisation drops.

Reply:

The JNJ compounds were dissolved in 100% DMSO as 10 mM (electrophysiology) or 50 mM (crystallography) stocks. As such, the final DMSO concentration in the crystallization drops was 0.2% and up to 0.1% in the functional experiments. Information about the solvent and compound stock concentrations has now been added to the respective sections in Methods (i.e., the Electrophysiology and Protein expression and purification sections).

Reviewer:

Typos: the authors keep putting no space between pH and its value, as in pH6.0 instead of pH 6.0

-p. 14 "Relatvie" -> Relative

-p. 25 What is the "viper diffusion method" (sic)? Probably vapour diffusion method?

-p. 26 Scordinger -> Schrodinger

Reply:

pHx.x has been changed to pH x.x now. Yes, we meant “vapour” and has now made the correction. All other typos have been corrected.

Responses to referee #3:

Reviewer:

A quantitative but still simple model would greatly improve the understanding of the experimental data and would (probably) be a convincing argument in the characterization of the binding site. I strongly suggest the authors to implement a quantitative kinetic model.

Reply:

This is an excellent suggestion! The qualitative kinetic model (old Fig. 5e) is now part of the new Fig. 6 (Fig. 6a), in which we have implemented a simple quantitative model. To test quantitatively whether our data necessitate the involvement of compound-bound desensitized states, we included in the new model (Fig. 6b) only minimal number of states (i.e., compound-bound and unbound closed states and compound-unbound desensitized states) and assumptions and explicitly excluded compound-bound desensitized states. This model produced good global fits to our kinetic data for JNJ-68969386, which strongly argues that our data can be fully accounted for without compound-bound desensitized states, agreeing quantitatively with our contention that desensitized channels are not compound bound at equilibrium. We now have added a corresponding paragraph each in the Results and Methods sections to describe the modeling work.

We agree with the reviewer that the quantitative treatment greatly improves the understanding of our experimental data and strengthens the arguments. We thank the reviewer for making this excellent suggestion.

Reviewer:

Do your binding pocket explain the drug effect on the pH dependence? I guess so, but I missed the explanation. Please, clarify.

Reply:

Yes, we touched on the structural basis of the drug effect on the pH dependence in the fourth paragraph in Discussion (in both the old and revised versions).

Reviewer:

Page 5: remove the sentence about preliminary data. Data not shown is not OK.

Reply:

The sentence is removed.

Reviewer:

Page 12: the relation between tachyphylaxis and desensitization is not clear to me. Try to explain this better.

Reply:

Tachyphylaxis is thought to result from channels that enter long-duration (slow) desensitized states, such that the interval between consecutive agonist pulses is not long enough to recover desensitized channels from these states. Such slow desensitized states are more likely induced by high-concentration protons, consistent with tachyphylaxis being more pronounced at lower pHs. We speculate that, by speeding up open-channel desensitization, our compounds decrease the likelihood of channels transitioning into such a slow desensitized state, hence retarding tachyphylaxis.

REVIEWERS' COMMENTS:

Reviewer #2 (Remarks to the Author):

I am fully satisfied by the answers of the Authors.
Paper can be accepted as it is.

Reviewer #3 (Remarks to the Author):

The authors have carefully responded to my questions. I am satesfied.